# The influence of detachment strength on the evolving deformational energy budget of physical accretionary prisms

Jessica McBeck[1, 2], Michele Cooke[1], Pauline Souloumiac[3], Bertrand Maillot[3], Baptiste Mary[3]

[1] Department of Geosciences, University of Massachusetts Amherst, MA, USA
[2] Now at: Physics of Geological Processes, The Njord Centre, Department of Geosciences, University of Oslo, Norway
[3] Département Géosciences et Environnement, Université de Cergy-Pontoise, Cergy-Pontoise, France

*Correspondence to*: Jessica McBeck (j.a.mcbeck@geo.uio.no)

**Abstract.** Tracking the evolution of the deformational energy budget within accretionary systems provides insight into the driving mechanisms that control fault development. To quantify the impact of these mechanisms on overall system efficiency, we estimate energy budget components as the first thrust fault pair develops in dry sand accretion experiments. We track energy budget components in experiments that include and exclude a basal layer of glass beads in order to investigate the influence of detachment strength on work partitioning. We use the measurements of normal force exerted on the backwall to estimate external work, and measurements of strain observed on the sides of the sandpacks to estimate the internal work, frictional work and work against gravity done within increments of each experiment. Thrust fault development reduces the incremental external work and incremental internal work, and increases the incremental frictional work and incremental gravitational work. The faults that develop within higher friction detachment experiments produce greater frictional work than the faults in experiments with glass bead detachments because the slip distribution along the detachments remain the same while the effective friction coefficient of the detachment differs between the experiments. The imbalance of the cumulative work budget suggests that additional deformational processes that are not fully captured in our measurements of the energy budget, such as acoustic energy, consume work within the deforming wedge.

## 1. Introduction

In accretionary prisms, margin-perpendicular convergence produces distributed layer-parallel compaction ahead of the wedge until strain localizes as slip along discrete frontal thrust faults (e.g., *Koyi*, 1995; *Burberry,* 2015; *Ghisetti et al.*, 2016; *Lathrop and Burberry*, 2017). The frictional strength of the basal detachment and rheology of the wedge control the internal deformation throughout the wedge prior to thrust fault development. Consequently, the timing and geometry of new fault development may vary with varying basal detachment strength.

Understanding the energetic tradeoffs between pervasive internal strain and fault development can help predict the timing and geometry of new frontal accretion thrust faults (e.g., *Del Castello and Cooke,* 2007; *McBeck et al.*, 2017). Investigations of the energy available for creating new fault surfaces show that new frontal accretion faults may only develop when the system has stored sufficient energy that can be expended in the development of new faults, which consume both acoustic and propagation energy (*Del Castello and Cooke,* 2007; *Herbert et al.,* 2015). Numerical simulations of physical accretion experiments demonstrate that optimizing the total external work done on the wedge can successfully predict the position and dip of new frontal accretion thrusts (*McBeck et al.*, 2017). These and other numerical efforts demonstrate the ability of energy criteria to successfully predict and understand fault development (e.g., *Masek and Duncan,* 1988; *Hardy et al.*, 1998; *Burbidge and Braun,* 2002; *Cubas et al.*, 2008; *Marshall et al.*, 2010; *Dempsey et al.*, 2012; *Mary et al.,* 2013; *Yagupsky et al.,* 2014; *Cooke and Madden,* 2014; *McBeck et al.*, 2016, 2017). However, few studies have used energy criteria to assess fault development within scaled physical analog experiments (e.g., *Ritter et al.*, 2018a). In particular, investigations of the energy budget of physical accretion experiments have thus far focused on two components: the external work and work of fault propagation (*Herbert et al.*, 2015). The complete deformational energy budget has not been calculated for deforming physical analog accretionary wedges. Calculating the components of the complete energy budget, including the energy consumed in frictional slip, uplift against gravity and internal diffuse deformation, provides a more complete picture of evolving energy partitioning throughout the system, and the tradeoffs between deformational processes associated with each work budget component.

We tracked the complete work budget from the onset of deformation through the formation of the first thrust pair within four scaled dry sand accretion experiments performed at the University of Cergy-Pontoise (UCP). Half of the experiments employed a basal layer of glass beads in order to provide a low friction basal detachment. Digital image correlation (DIC) and force measurements enabled estimation of the energy expended in uplift against gravity, work done against frictional slip, energy stored as pervasive off-fault deformation, and the total external work done on the physical accretion experiments. We compare the evolution of work budget partitioning for experiments with high and low frictional detachments in order to reveal the sensitivity of different processes captured in the work budget on detachment strength.

## 2. Background

Field observations, physical analog experiments and numerical simulations inform the energy budget associated with the spatiotemporal evolution of faults within accretionary systems.

### 2.1. Energy budget of accretionary systems

The material properties that control strain localization and thrust fault development in accretionary prisms include the evolving frictional strength of the detachment, the internal frictional coefficient and cohesion of the relatively intact material within the wedge, the evolving cohesion and frictional strength of the thrust faults, and the effective stiffness of the wedge (e.g., *Davis et al.*, 1984; *Dahlen et al.*, 1984; *Lohrmann et al.*, 2003; *Le Pourhiet*, 2013). An energy budget framework can quantify the impact of each of these evolving properties, and the resulting stresses and strains within the wedge, on system deformation (e.g., *Dahlen*, 1988; *Burbidge & Braun*, 2002). In particular, recent investigations of fracture propagation and interaction indicate that tracking the evolution of various components of the work budget of a deforming fault system sheds insight into mechanical processes that control fault system development (e.g., *Del Castello and Cooke*, 2007; *Dempsey et al.*, 2012; *Cooke and Madden*, 2014; *Herbert et al.*, 2015; *McBeck et al.*, 2017; *Newman & Griffith*, 2014). The energy budget of a deforming fault system includes the internal work of deformation, $W_{int}$, the work of uplift against gravity, $W_{grav}$, the work done against frictional sliding along fractures, $W_{fric}$, the work required to create new fracture surfaces, $W_{prop}$, and the work released as seismic energy, $W_{seis}$. The external work, $W_{ext}$, quantifies overall system efficiency and is the sum of the product of the normal and shear displacements and tractions on system boundaries.

The evolving energy budget of accretionary systems can help predict the spatiotemporal development of accretion faults (*Burbidge & Braun*, 2002; *Del Castello and Cooke*, 2007; *McBeck et al.*, 2017). *Del Castello and Cooke* (2007) show that accretionary faults evolve to optimize the total work done in the system, even as individual components of the work budget increase. In these numerical accretion simulations, the development of the new forethrust increases $W_{int}$, but decreases $W_{fric}$ by a greater degree, which correspondingly decreases $W_{ext}$ (*Del Castello and Cooke*, 2007). *Del Castello and Cooke* (2007) propose that a fault will propagate only when the gain in efficiency exceeds the energetic cost ($W_{prop} + W_{seis}$) of propagating the fault. Analysis of the change in external work associated with the propagation of new faults in physical accretionary wedges indicate that greater energy per fault area ($W_{prop}$) is expended during fault propagation in thicker sandpacks and higher frictional strength sand (*Herbert et al.*, 2015). *McBeck et al.* (2017) demonstrate that the forethrust geometry that optimizes the overall efficiency of the fault system, $W_{ext}$, closely matches the position and dip of forethrusts observed in physical accretion experiments. The success of using external work optimization to assess the geometry and timing of accretion thrust development suggests that an analysis of the entire work budget throughout the transition from diffuse deformation to localized faulting may provide insight into the processes associated with this transition.

In this study, we track the evolving energy budget in wedges with detachments of differing strengths in order to assess how detachment strength influences the overall efficiency of the wedge (external work), as well as the magnitude of fault slip (frictional work), uplift against gravity (gravitational work) and pervasive internal strain (internal work). We expect that higher detachment strength may 1) increase external work by providing greater resistance to contraction, 2) increase the frictional work if the magnitude and distribution of slip along the detachments remain the same and 3) may increase the internal work prior to thrust faulting due to greater shear tractions on the detachment that produce higher stresses within the wedge.

**2.2. Onset of strain localization in accretionary systems**

Field observations suggest that off-fault deformation within accretionary prisms, such as distributed microcracking and ductile deformation, produce diffuse thickening of strata that accommodates a significant proportion (30-40%) of total shortening (e.g., *Lundberg & Moore*, 1986; *Morgan et al.*, 1994; *Morgan & Karig*, 1995; *Moore et al.*, 2011). Recent retro-deformations of seismic

profiles from the central Hikurangi Margin suggest that macroscopic thrust faults and folds accommodate <50% of the margin-perpendicular shortening due to plate convergence (*Ghisetti et al.*, 2016). The protothrust zone identified in seismic transects of the Nankai wedge between the trench and a zone of imbricate thrust faults (e.g., *Moore et al.*, 2011) indicates that diffuse compaction within crustal wedges may be concentrated into particular zones, and not homogeneously distributed throughout the wedge.

The million to thousand year intervals between faulting events within crustal accretionary wedges amplifies the need of scaled and quantitatively-monitored physical experiments to directly document the transition from diffuse deformation (i.e. greater internal work) to localized faulting (i.e. greater frictional work). Recent analytical analyses indicate that the solutions governing the stability and critical taper of non-cohesive accretionary wedges do not differ in subaerial (dry) and submarine (normally pressured) conditions (*Lehner & Schöpfer*, 2018), suggesting that insights from dry physical accretion experiments are applicable to 105 submarine, crustal accretionary prisms. Physical accretion experiments reveal distributed layer-parallel shortening associated with detachment slip prior to the localization of slip along thrust faults (e.g., *Mulugeta & Koyi*, 1992; *Koyi*, 1995; *Burberry,* 2015; *Lathrop & Burberry*, 2017; *McBeck et al.*, 2017). In dry sand accretionary wedges, inclusion of a ductile detachment greatly increases the overall distributed internal strain associated with folding and thrusting (*Lathrop & Burberry*, 2017). Incremental displacement fields of the sides of wedges indicate that short-lived shear bands episodically develop until strain localizes onto a 110 single frontal forethrust (e.g., *Bernard et al.*, 2007; *Dotare et al.*, 2016). The short-lived shear bands contribute to distributed internal deformation prior to localized slip.

Previous experiments observed transient increases and decreases in external force associated with fault development, indicative of varying magnitudes of external work (*Nieuwland et al.*, 2010; *Cruz et al.*, 2010; *Souloumiac et al.*, 2012; *Herbert et al.*, 2015; *Ritter et al.*, 2018b). Similarly, the partitioning of the energy budget is expected to differ before and after thrust fault development. 115 Prior to faulting, distributed internal strain ($W_{int}$) may accommodate a larger percentage of the overall work budget than after thrust fault development, when both internal work and localized frictional slip on faults ($W_{fric}$) accommodate the applied deformation. The strength of the detachment is expected to influence energy budget partitioning. In wedges with weaker detachments, frictional work and gravitational work should consume lower portions of the energy budget, whereas internal work may consume a higher portion of the energy budget compared to wedges with stronger detachments.

**3. Methods**

To gain insight into the partitioning of energy components during fault development, we use four accretion experiments performed at the University of Cergy-Pontoise (UCP). Here, we describe the setup and quantitative post-processing of the physical experiments. Then, we describe the method of constraining the effective stiffness of the wedges and the calculation of the work budget components using observations from the experiments.

**3.1. Design of accretion experiments**

In these experiments, an electric screw motor translated one wall of the experimental apparatus toward the other in order to contract the dry sand wedge. Throughout the duration of each experiment, three cameras captured photos of different views of the sandpack at one second intervals (Fig. 1). One camera took photos of the top of the sandpack while the other two cameras took photos of the

sandpack cross sections through both of the glass sidewalls. In addition, 16 uniaxial strain gauges mounted on the backwall of the experimental apparatus recorded changes in normal force exerted on the backwall throughout the experiment. Previous physical experiments performed at UCP indicate that variations in normal force on the backwall coincide with fault development (e.g., *Souloumiac et al.*, 2012; *Herbert et al.*, 2015).

The sand deposition method strongly controls the density and frictional properties of the system, and thus how the sandpack accommodates strain (e.g., *Krantz,* 1991; *Lohrmann et al.,* 2003; *Maillot*, 2013). To construct homogeneous and isotropic sandpacks, a sedimentation device, designed and built at UCP, sieves the sand two or three times before deposition (*Maillot*, 2013). We used this sedimentation device to construct the sandpacks.

To investigate the impact of a weaker basal detachment on the energy budget, two of the experiments included a 0.5 cm thick layer of glass beads (median grain size of 160 μm), while the other two experiments did not include this layer. In the experiments including the basal glass bead layer, a 4 cm thick layer of CV32 sand overlaid a 0.5 cm thick layer of glass beads. In the experiments excluding the glass bead layer, the initial sandpack had 4 cm thickness. For both experiments, we added a protowedge of sand on top of the rectangular sandpack to concentrate deformation away from the backwall of the experimental apparatus. The protowedge slope was the angle of repose of the sand (30°), and the base of the protowedge was 10 cm long.

For each set of two experiments that either include or exclude a glass bead layer, we modified the configuration of the experimental apparatus in order to remove bias arising from sidewall friction following the methods of *Souloumiac et al.* (2012). Frictional sliding of the sandpack along the glass sidewalls produces a net force vector that has a different direction depending on whether the base is fixed or moving. The drag of the sidewall produces curvature in the strike of thrusts that is expected to develop concave to the motor direction. We changed the direction of the sidewall friction drag by switching the position of the backwall with respect to the motor (Fig. 1). We aim to eliminate the bias in the experimental observations that arises from sidewall friction by synthesizing observations from the two experiments.

In two experiments (E373, E374), the base of the apparatus slid under the fixed backwall (Fig. 1C). In the other experiments (E375, E376), the base was fixed and the screw motor pushed the backwall so that the sand slid over the stable glass base (Fig. 1D). In the remaining text, we refer to these different configurations as the *moving base* (E373, E374) and *moving backwall* (E375, E376) experiments. The opposing apparatus configurations used here were expected to produce only minimal differences in the position of the thrusts at the sandpack sides because the ratio of the area of the sandpack in contact with the sidewalls to the area in contact with the base was low (~8/100) in these experiments. *Souloumiac et al.* (2012) find that when this ratio is <0.1, the influence of sidewall friction produces only small changes in the thrust geometry along-strike. The experiments of this study will confirm the calculations of *Souloumiac et al.* (2012) if the opposing configurations produce minimal changes in the position of the thrust faults at the sandpack sides. If the work budgets of the experiments with differing configurations also remains similar, then this study will provide further confirmation that the apparatus configuration has a minimal impact on strain partitioning.

## 3.2. Incremental displacement and strain fields

To constrain the active fault geometry and calculate the energy budget components throughout each experiment, we used digital image correlation (DIC) to calculate the incremental displacement fields of the top and sides of the sandpack (Fig. 2). For each

sequential pair of photos, DIC analysis determines the incremental displacement field at a grid of points via DIC of pixel constellations (e.g., *Adam et al.*, 2005). The incremental displacement fields revealed active thrusts where slip produces sharp gradients in the displacement field (i.e., localized shear and normal strain) (Fig. 2). We calculated the incremental shear strain fields of each of the views of the sandpack using the curl of the incremental displacement field, which indicates the incremental rotational angle. In our analysis, the tangent of the incremental rotation angle approximately equals the angle (in radians) because we investigate small changes in displacement. Thus, the curl of the incremental displacement field approximately equals the incremental shear strain under these conditions. This calculation of shear strain is independent of coordinate system, so that shear along all orientations have equal representation.

To perform DIC, we employed several techniques to produce accurate incremental displacements from the photos. During the experiment, we acquired 10 MB color 24-bit JPEG photographs. We orthorectified the photos to remove distortion near the edges, converted the color photos to gray scale and increased the contrast in order to aid the DIC calculation. We performed these steps, and performed DIC with a triple pass filter, using the PIVlab MATLAB™ plug in (*Thielicke*, 2018). The analysis produced incremental displacements with a spatial resolution of 0.01 cm, and temporal resolution of 2 seconds, or 0.2 mm of applied backwall displacement.

### 3.3. Force measurements

To capture fluctuations in normal force exerted on the backwall throughout each experiment arising from strain hardening and softening of the sandpack, we used 16 uniaxial strain gauges mounted on plaques in contact with the backwall (blue in Fig. 1C-D). Four plaques each containing four uniaxial strain gauges were connected in series so that the mean strain of the 16 sensors are reported (Fig. S1). Throughout each experiment, uniaxial strain was recorded at 0.1 second intervals. Calibration of the strain gauges to known weights produced a linear relationship of machine strain units to applied weights with $R^2$=0.99 (Fig. S1). We removed noise from the force data with a standard median filter of 30 measurements. This filter of 0.030 mm increments of applied motor displacement removed noise without over-smoothing the peaks and troughs of the force curve.

### 3.4. Estimation of energy budget components

Each of the components of the work budget arise from integrating the product of force and displacement over the corresponding deforming region (e.g., *Cooke and Madden*, 2014). We estimated the incremental gravitational, frictional, internal and external work at stages of the experiments using the incremental displacement fields of the sides of the wedge and the normal force exerted on the physical backwall. Previous work presents the equations for each component of the energy budget (i.e., *Cooke and Madden,* 2014). Here, we describe how we used the experimental measurements to estimate the values used in those equations.

At the slow stepper motor rate prescribed here (0.1 mm/s), the true motor displacement rate appears to fluctuate within 1 second intervals. We observed this fluctuation in the screw of the screw motor, and in the varying incremental displacements calculated with DIC. Consequently, this fluctuation impacted both the normal force exerted on and displacement of the backwall. We do not observe stepper motor rate fluctuations at time intervals >1 second. Therefore, to minimize impact of non-steady incremental backwall movement recorded in the 1 second photo intervals, we stacked sequential incremental displacement fields, producing individual incremental displacement fields representing 0.2 mm of applied displacement over 2 seconds. To track the elapsed time

in each experiment, and calculate the external work, we extracted the total backwall displacement from the DIC displacement fields, rather than rely on the prescribed motor rate and elapsed time.

The motor supplies work to the wedge by applying a force to the moving wall as it displaces at the prescribed velocity. The cumulative external work done from the start of deformation to the stage of interest is the area under the force-displacement curve (e.g., *Cooke and Murphy*, 2004). Thus, the external work done in an increment of an experiment may be considered to be the area under the force-displacement curve for that increment of backwall displacement measured from the DIC fields of the top view. To calculate the external work, $W_{ext}$, done in increments of the physical experiment, we used the backwall displacement within each increment measured from the DIC and the force exerted on the backwall. Assuming that the change in force over the increment is linear,

$$\partial W_{ext} = U_n F_n^0 + \frac{1}{2} U_n (F_n^1 - F_n^0) \tag{1}$$

where $U_n$ is the normal displacement within the increment of the experiment, $F_n^0$ is the force at the beginning of the increment, and $F_n^1$ is the force at the end of the increment.

The work against gravity, $W_{grav}$, measures the work of uplift against gravity, and depends on the vertical normal stress and uplift. We used measurements of the density of sandpack produced with the UCP distributor (*Maillot*, 2013) to estimate the vertical normal stress due to lithostatic compression. We integrated the product of this normal compression and the vertical displacement fields of the sandpack sides over the deforming region to estimate $W_{grav}$. We integrated this product over the 1 m width of the sandpack in order to calculate the volumetric $W_{grav}$ throughout the wedge.

The frictional work, $W_{fric}$, done along faults in the experiment is the area integral of the product of slip along faults and the shear traction on the faults along fault surfaces. We estimated the slip along the detachment fault using the horizontal velocity of the sand at the base of the physical experiment calculated through DIC. We constrained the slip along the backthrust and forethrust by integrating the shear strain field near the observed thrusts. We first identified areas of localized slip within the sandpack as locations with incremental shear strain above a critical value (0.0005 in the approximately 0.2 mm displacement increment). We integrated the product of shear strain and shear traction within these regions of high shear strain to calculate $W_{fric}$. To estimate shear traction associated with slip, we used the Coulomb criterion for frictional slip. Assuming no cohesion of the sand grains, the shear traction on the faults is the normal traction multiplied by the sliding friction. We assume lithostatic normal tractions on the thrust and detachment faults. Following laboratory measurements of basal friction of sand and glass beads (*Klinkmüller et al.*, 2016), we used sliding frictions of 0.3 and 0.2 for the detachments that develop within sand and glass beads, respectively. Following laboratory measurements of the effective friction of faults that develop within sandpacks produced by the UCP depositor (*Maillot*, 2013), we used 0.7 for the thrust fault sliding friction. We integrated the product of slip, normal stress and effective friction along the area of the fault surfaces throughout the sand pack in order to calculate $W_{fric}$.

Internal work measures the energy consumed in internal strain of the material as quantified by strain energy density, which is a product of the components of the stress and strain tensors. Here, we use this quantity to capture the elastic contribution to deformation outside the fault zones. Whereas localized faults within the sandpack deform inelastically, the sandpack around the faults stores and releases strain in concert with fault slip. The overall elastic response of the sandpack produces the observed

reduction in external force associated with the growth of new thrusts (Fig. 3A). To estimate the internal work, $W_{int}$, done via distributed deformation in the sandbox outside the fault zones, we implemented Hooke's elasticity laws in order to estimate the incremental stress from the observed incremental strain field. The product of this approximated stress and measured strain field forms the strain energy density, which we integrated through the volume of the sandbox to determine the internal work (e.g., *Cooke and Madden*, 2014). We first integrated the strain field for each of the components of the strain tensor over the area of the deforming region that does not include the fault zones of localized slip (where shear strain exceeds a critical value). Multiplying these values by 1 m gives the strain integrated throughout the volume of the sandpack. Next, we converted the integrated strain tensor components to approximate integrated stress tensor components using Hooke's law for plane strain, which utilize the material properties Poisson's ratio and Young's modulus. We set Poisson's ratio to 0.2, which is within the range of dry sand values from 0.2-0.4 (*Gercek*, 2007). We used our estimates of the effective stiffness of these sandpack for Young's modulus (Sect. 3.5, Fig. S2).

Throughout each experiment, the gravitational, frictional and internal work calculated from the displacement field of one side of the sandpack may differ from the work calculated from the opposite side of the sandpack. These differences arise from changes in the fault geometry along strike, as well as in the quality of the incremental displacements calculated through DIC. Consequently, we present the components calculated from the incremental displacement fields of both sides of each experiment. This demonstrates the variability that can be expected in different accretion experiments (e.g., *Cubas et al.*, 2010).

### 3.5. Constraining effective stiffness with force and strain

The calculation of internal work within the accretion experiment depends on the stress state within the wedge, which is not measured in these experiments. To constrain the internal stress state from the observed strain fields using Hooke's laws and the effective stiffness of the material, we must estimate the effective stiffness of the sand wedges. To do this estimation, we used measurements of the normal force exerted on the backwall and contraction along the sides of the sandpack (Text S1). The slope of the resulting stress-strain curve at 50% of the peak stress reveals the effective stiffness of the sand (e.g., *Barber*, 2002). Like other geotechnical materials with overall non-linear rheology, sand aggregates show linear behavior around 50% of peak stress (e.g., *Klinkmüller et al.*, 2016). Consequently, the slope of the normal stress-normal strain curve preceding thrust fault development provides an acceptable estimate of the stiffness of the wedge to use in the approximations of internal work.

We calculated the tangent of the horizontal longitudinal strain, $\varepsilon_{xx}$, and normal stress on the backwall, $\sigma_n$, over an interval of 1 mm of backwall displacement at 50% peak stress (Fig. S2). Each sandpack was built of the same material, and with the same deposition technique, and so each wedge was expected to have similar effective stiffness regardless of apparatus configurations. Nevertheless, the sandpacks in the moving backwall experiments have slightly higher estimates of effective stiffness (1.2, 1.3 MPa) than the sandpack of the moving base experiments (0.6, 0.9 MPa). Because the evolution of force is similar preceding the peak force in each of the experiments (Fig. 3), differences in the effective stiffness estimates arise from local variations in the sandpack that impact the incremental horizontal displacement fields. To estimate internal work in Fig. 4 and Fig. 8, we used the mean of the four stiffness measurements (1 MPa). The supplemental information (Fig. S5) considers the influence of using different values of effective stiffness in the calculation of internal work.

### 3.6. Kinematic compatibility assessment

The accuracy of the incremental displacement field calculated through DIC varied due to slight differences in illumination, focal depth and camera positioning between each experiment, and between each side of one experiment. To assess the robustness of each incremental displacement field, we calculated the 2D kinematic compatibility of each field. We solved the 2D kinematic compatibility equation at all specified points from the strain, $\varepsilon_{ij}$, as

$$270 \quad K = \frac{\partial^2 \epsilon_{11}}{\partial x_2^2} + \frac{\partial^2 \epsilon_{22}}{\partial x_1^2} - 2\frac{\partial^2 \epsilon_{12}}{\partial x_1 \partial x_2} \qquad (2)$$

Robust incremental displacement fields will have no apparent movement of material in and out of the plane due to artificial noise, so that the kinematic compatibility, $K$, will be zero throughout the field. To derive a representative measure of the artificial noise level throughout each displacement field, we find the mean plus one standard deviation of the $K$ field, $K^F$ (Fig. S3). We considered displacement fields with $K^F$ above one standard deviation from the mean of the $K^F$ of all the fields as unreliable, and we removed
these displacement fields from the calculation of the work budget components. Most of the incremental displacement fields of the experiments contained an acceptable amount of artificial noise (Fig. S3D). The incremental displacement fields derived from the photos taken from the camera on the right side of experiment E375 had the highest noise levels, and consequently most of these incremental displacement fields were excluded from the work budget analysis.

## 4. Results

First, we describe the relationship between thrust fault development and the overall efficiency of the wedges. Then, we use the experimental data to estimate the changes to the energy budget associated with the development of new accretionary faults. Finally, we link specific deformational processes to the evolution of the work budget components.

### 4.1. Relationship of external work and faulting events

In the four experiments, the sequence of faulting followed that observed by *Herbert et al.* (2015). The DIC strain fields reveal that
first, a backthrust-forethrust pair developed near the leading edge of the protowedge (Fig. 2, Fig. 3). Then, a new backthrust propagated in front of the pre-existing backthrust, further from the backwall. Next, a new forethrust propagated in front of the pre-existing forethrust. Finally, a new backthrust-forethrust pair developed in front of the latest forethrust.

The evolution of the normal force exerted on the backwall highlights these faulting events (Fig. 3). The measured backwall force tracks the evolution of external work because the external work is a function of the normal force exerted on the backwall and the
290 applied backwall displacement. For each experiment, the normal force rose to a local maximum (peak) and fell to a local minimum (trough) coincident with new fault development (Fig. 3). Prior to thrust faulting, the backwall force steadily increased as the wedge compacted and thickened, and greater work was required to maintain the backwall velocity. The three different camera views of the physical experiment enabled identification of thrust faulting events at the top of the sandpack and at both sides of the experiment (Fig. 2). The faults, identified as localized shear strain in the side views and normal contraction in the top view, emerged at the top
of the sandpack 0.5-1 mm before the force peak as the force curve transitions from linear to the strain-softening behavior (Fig. 3B-E). The force trough occurred after we observed the first evidence of faulting on either side of the experiment, confirming the findings of laboratory rheometric tests that faults within sand continue to weaken with decreasing friction coefficient as they accrue several millimeters of slip (e.g., *Vermeer*, 1990; *Lohrmann et al.*, 2003; *Ritter et al.*, 2016). Consistent with findings of *Herbert et*

*al.* (2015), the experiments with glass bead detachments had lower force drops (24-28 N) than the experiments with sand detachments (30-36 N) (Fig. 3A). This result indicates that thrust fault development in wedges with stronger detachments produces greater reduction in $W_{ext}$. Consequently, new fault growth in wedges with stronger detachments is associated with greater increases in mechanical efficiency. Following the force trough, the backwall normal force remained relatively constant in the experiments with weaker detachments, whereas the force began to rise in the experiments with stronger detachments (Fig. 3A). This behavior suggests that wedges with weaker detachments provide less resistance to the continued backwall displacement and so have greater mechanical efficiency than wedges with stronger detachments following faulting.

### 4.2. Energy budget evolution

To determine the contribution of each deformational process to the increase in overall efficiency associated with fault development, we estimated the energy consumed or stored in increments of applied displacement throughout each experiment. First, we calculated each incremental work budget component over successive increments of backwall displacement (Fig. 4). This incremental work corresponds to the rate of work.

Thrust faulting decreased the incremental $W_{ext}$ and incremental $W_{int}$, and increased the incremental $W_{fric}$ and incremental $W_{grav}$ (Fig. 4). The incremental external work, $\delta W_{ext}$, rose from the onset of displacement to ~35 mJ, and then began to fall following thrust fault development (Fig. 4A). Thrust faulting decreased $\delta W_{ext}$ by ~10 mJ, or by 20% of its value preceding faulting. This decrease corresponds to the onset of strain softening as the backwall force curve deviated from its initial linear trend at the onset of loading.

The incremental internal work, $\delta W_{int}$, was elevated preceding thrust fault development, and fell as the thrusts develop (Fig. 4B). Thrust faulting decreased $\delta W_{int}$ by about 0.02 mJ, or by 20-90% of its value preceding faulting. For each experiment, the largest rate of increasing off-fault deformation occurred immediately preceding thrust fault development (Fig. 4B). Experiment E375 most clearly shows this relationship relative to the other experiments. The period of thrust fault growth had a decreasing rate of off-fault deformation accumulation ($\delta W_{int}$) as strain localized onto the faults, rather than accumulating in the surrounding material.

Prior to thrust fault development, work against frictional slip occurred along the basal detachment fault. For each experiment, thrust fault development increased $\delta W_{fric}$ from the values associated with basal detachment slip preceding thrust faulting (Fig. 4C). The growth of the thrust faults produced greater increases in $W_{fric}$ in the sand detachment experiments than the glass detachment experiments. The slip distribution along the detachments are similar, so this difference in $W_{fric}$ arose **primarily** from the greater friction coefficient of the sand detachments (0.3) relative to the glass detachments (0.2) (Fig. S4). After the thrust faults were fully formed (6 mm of backwall displacement), the cumulative slip distribution along the backthrust in experiment E375 (sand detachment) had greater slip than experiment E374 (glass detachment) (Fig. S4), while the forethrust had less slip in the sand detachment experiment. However, the slip distribution along the detachments in both experiments were similar, indicating that the differences in $W_{fric}$ arose from the differing friction coefficients. The difference in the effective friction coefficient of the detachment in the sand (0.3) and glass (0.2) detachment experiment lead to larger detachment contributions to $W_{fric}$ in the sand (256 mJ) compared to the glass (166 mJ) detachment experiment.

Thrust faulting increased $\delta W_{grav}$ (Fig. 4D). The $\delta W_{grav}$ began to increase prior to thrust fault development as the propagation of the detachment from the backwall produced diffuse uplift. The general increase in $\delta W_{grav}$ coincident with thrusting reflected the

increased uplift rate associated with slip along the new thrusts. Slip along the thrust faults more efficiently thickened the sandpack than the prefaulting diffuse processes.

Prior to faulting, the increasing external work applied to the wedge was accommodated with increasing internal work, increasing uplift against gravity, and increasing work against frictional sliding along the detachment. The maximum rate of increase in frictional and gravitational work budget components correlates with the minimum incremental external work, minimum rate of increasing internal work and the peak in backwall normal force. After this peak force, the system fails and more work was expended as work against frictional slip and stored as work of uplift against gravity, than stored as internal work. Of the calculated
components of the work budget, thrust faulting produced the smallest changes in $\delta W_{int}$, and largest changes in $\delta W_{fric}$.

### 4.3. Linking the energy budget to deformational processes

The positive values in the incremental energy budgets suggest that the cumulative work components monotonically increase throughout thrust fault development, but at varying rates. The incremental energy budget of the physical experiments indicates that thrust fault development increased $\delta W_{fric}$ and $\delta W_{grav}$, while decreasing $\delta W_{int}$ and $\delta W_{ext}$. With localized slip along the new thrust
faults, displacement of the backwall produced smaller increases in the backwall normal force following thrust fault development than preceding development, reducing $\delta W_{ext}$, and so increasing the mechanical efficiency of the system. Prior to thrust faulting, the applied strain was less efficiently accommodated as distributed off-fault deformation, $\delta W_{int}$. Because internal work is a conservative term, the decrease in $\delta W_{int}$ associated with thrust fault development suggests that work stored as internal strain before faulting was expended as other forms of work when fault slip localizes (i.e., $\delta W_{fric}$ and $\delta W_{grav}$).

The vertical displacement fields of the sides of the physical sandpacks demonstrate that diffuse uplift preceded the development of discrete thrusts (Fig. 5). The evolution of $\delta W_{grav}$ (Fig. 4) reflects the broad uplift preceding thrust development, as well as the localized uplift following thrust development. Prior to thrust fault growth, the broad region of uplift developed near the distal tip of the detachment (Fig. 5A), increasing $\delta W_{grav}$, and then as slip localized along thrust faults (Fig. 5B), $\delta W_{grav}$ remained stable. After thrust fault formation, $\delta W_{grav}$ remained elevated at ~3 mJ relative to the prefaulting stage (Fig. 4D) because slip along the thrust
faults continued to promote uplift (Fig. 5B). The more effective uplift against gravity associated with slip along the new thrust faults increased $\delta W_{grav}$ when the thrust faults first develop, and then continued slip along the thrust faults produced similar $\delta W_{grav}$ in subsequent stages.

The increase in $\delta W_{fric}$ following thrust fault development (Fig. 6) illuminates the influence of thrust slip partitioning on the accretion experiments energetic evolution. Thrust development increased the total active fault area and fault slip in the system, increasing
$\delta W_{fric}$. Thrust faulting in the experiments with the sand detachments produced greater $\delta W_{fric}$ than in the experiments with the glass detachment (Fig. 5C) because the higher friction detachment faults hosted similar slip to the lower friction detachment faults (Fig. S4). The shear strain of the incremental displacement fields of the physical experiment excluding the glass bead layer (E375) show localized slip on the backthrust at the force trough of the experiment (Fig. 6A). However, the experiment including the glass bead layer (E374) produced lower shear strain in the backthrust region at the force trough (Fig. 6B). The entrainment of glass beads at
the root of the forethrust promotes slip on the forethrust at the expense of the backthrust in the glass bead experiments (Fig. 6C, Fig. S4).

Tracking the evolution of the strain tensor components provides insight into internal deformation prior to thrust fault development, which contributes to the internal work of the system (Fig. 7). Preceding faulting in experiment E375, the wedge contracted horizontally and thickened vertically, producing high normal strains dispersed through the wedge, and localized shear strain along the detachment (Fig. 7A-B). With continued backwall displacement, horizontal contraction and vertical extension shifted toward the distal tip of the detachment, where the thrusts ultimately developed, and the zone of high shear strain surrounding the detachment extended further from the backwall. In the incipient stages of thrust development, horizontal contraction and vertical extension concentrated along the incipient thrusts, forming broad bands of high strain a few centimeters thick (Fig. 7C). As the thrust faults slipped, the zones of high horizontal contraction and vertical thickening localized, and shear strain along the forethrust increased while shear strain along the backthrust remained low (Fig. 7D).

## 5.  Discussion

Synthesizing quantitative observations from physical experiments reveals energy partitioning associated with thrust fault development. Here, we discuss how the energy budget balanced in these experiments, and the impact of the apparatus design on strain localization.

### 5.1.  Balancing the work budget

Evolving diffuse deformation, uplift and localizing strain change the overall work done in the system (Fig. 8). Following the conservation of energy, the total $W_{ext}$ should equal the sum of the work budget components (e.g., *Cooke and Madden*, 2014),

$$W_{ext} = W_{prop} + W_{seis} + W_{int} + W_{grav} + W_{fric} \tag{3}$$

However, in each experiment, the sum of $W_{int}$, $W_{fric}$ and $W_{grav}$ are 100-200 mJ less than $W_{ext}$ after the faults develop, at 6 mm of backwall displacement (Fig. 8), or 20-40% of $W_{ext}$. Here, we focus on experiment E376 because its displacement fields have less noise than the other experiments (Fig. S3), but each experiment produces this deficit in the cumulative work budget. Underestimates of $W_{int}$ may arise from errors in estimates of the effective sandpack stiffness. The effective stiffness may be larger at later stages of the experiment after the sand has compacted under horizontal contraction (e.g., *Koyi*, 1995). In addition, the accommodation of localized slip along thrust faults produces more compliant wedges that require less normal force on the backwall with additional deformation (e.g., *McBeck et al*., 2017). In addition to these temporal variations, the effective stiffness of experimental and crustal wedges will also vary spatially as less dense material in front of the wedge has lower effective stiffness than compacted material within the wedge (e.g., *Wang & Hu*, 2006). Underestimates of $W_{fric}$ may arise from underestimates of the slip along the faults and/or normal tractions acting on the faults. We assume lithostatic compression along the faults. The overall horizontal shortening may produce compressive normal tractions greater than lithostatic along dipping thrust faults. Underestimates of $W_{grav}$ may arise from underestimating the effective density of the sandpack as the wedge evolves and compacts.

To constrain the potential range in error in $W_{int}$ that arises from the chosen effective stiffness, we calculate the incremental $W_{int}$ through experiment E376 using effective stiffnesses ranging from 0.25 to 100 MPa (Fig. S5). Increasing the stiffness by two orders of magnitude from 1 MPa to 100 MPa increases the average $\delta W_{int}$ done throughout the experiment from 0.01 mJ to 0.5–1.3 mJ for

the opposite sides of the sandbox. The maximum range of these estimates remain lower than the average $\delta W_{fric}$ (4-10 mJ) and $\delta W_{grav}$ (1.5-3 mJ), confirming that $\delta W_{int}$ consumes less work than the other components of the energy budget.

To constrain the potential range in error in $W_{fric}$ produced by assuming lithostatic normal compression on the thrust faults, we calculate the $\delta W_{fric}$ using differing ratios of the lithostatic stress, $\sigma_2$, and horizontal normal compression that arises from tectonic forces and burial depth, $\sigma_1$ (Fig. S5). Faults with a coefficient of internal friction of 0.6 and low inherent shear strength will form by Coulomb failure at $\sigma_1/\sigma_2$ ratios less than 3 in the deepest portion of these sandpacks (4 cm of 1700 kg/m$^3$ density sandpack, $\sigma_2 = 0.6$ kPa). Increasing the ratio of $\sigma_1/\sigma_2$ from 1 to 3 increases the average $\delta W_{fric}$ throughout experiment E376 from 4 mJ to 6 mJ. Consequently, the true average $\delta W_{fric}$ may be closer to 6 mJ (when $\sigma_1/\sigma_2$=3) than 4 mJ (when $\sigma_1/\sigma_2 = 1$) in experiment E376, reducing the apparent energy budget deficit.

In our calculations of $\delta W_{fric}$, we did not incorporate the contribution of sidewall friction. However, the geometry of the thrust faults suggests that our liberal application of Rain-X to the glasswalls produced surfaces with low friction coefficients. The lack of pronounced curvature of the strike of the thrust faults viewed from the top of the sandpack (Fig. 2, Fig. S5) indicates a minimal impact of sidewall friction, and thus a low sidewall friction coefficient.

Noise in the displacement fields may also have introduced errors into the work budget calculations. However, variations between the true displacement field and calculated displacement field are likely smaller than differences in the displacement fields on different sides of each experiment, where differences in fault geometry influence the displacement fields. Although we do not know the displacements within the center of the sandpack, the record of displacement at the two side windows provides two viable expressions of the experiment's overall deformation. Consequently, we consider the difference in the work budget values calculated for either side as estimates of uncertainty in the calculations. The shaded regions in Fig. 8 show the difference in the values calculated for each side, and function as uncertainty estimates.

The imbalance of the work budget suggests that other processes besides diffuse internal deformation, frictional slip and uplift against gravity consume work. Processes not directly considered in this study include the creation of new fault surfaces, and the generation of seismic energy or acoustic emissions (e.g., *Madden et al.,* 2017; *Herbert et al.,* 2015). Previous investigations of physical experiments indicate that the total work of creating new fault surfaces and released in seismic energy during fault propagation ($W_{prop} + W_{seis} = W_{grow}$) for sandpacks with 3 cm thickness is 385±150 mJ per each m$^2$ of new fault area (e.g., *Herbert et al.,* 2015). Using this relationship and the area of the faults observed in experiment E374, the cumulative $W_{grow}$ after the thrust faults develop is 130±50 mJ, which could account for most but not all of the 200 mJ of work budget deficit (Fig. 8).

### 5.2. Comparison to numerical and crustal accretionary prisms

In these experiments, we find that $W_{fric}$ consumes the largest portion of the energy budget, while $W_{grav}$ and $W_{int}$ consume the smallest portions. After the formation of the first thrust pair, the cumulative $W_{fric}$, $W_{grav}$, and $W_{int}$ consume 65-80%, 15-20%, and 2-5% of the cumulative $W_{ext}$, respectively. In a seminal analytical analysis of the mechanical energy budget, *Dahlen* (1988) found a similar partitioning of the energy budget using a critical wedge theory analysis of the fold-and-thrust belt of western Taiwan, with 45% in $W_{fric}$ along the basal detachment, 39% in $W_{grav}$ and 16% in internal dissipation. However, numerical models of physical accretion experiments that use elastic rheology show a different partitioning of the work budget than documented in these experiments and

by *Dahlen* (1988). For example, the elastic models show a larger proportion of internal work to gravitational work (*Del Castello and Cooke*, 2007). In these elastic models, $W_{grav}$ consumes a smaller portion (<10%) of the energy budget than $W_{int}$ (20-30%), and $W_{fric}$ consumes the largest portion (60-70%) throughout an accretion-underthrusting cycle (*Del Castello and Cooke*, 2007). These differences may arise because the sand aggregate deforms as an elastic-perfectly plastic material, whereas individual sand grains deform elastically. Consequently, at the onset of horizontal contraction in the experimental wedges, the sand aggregate compacts, producing stored internal work, but further horizontal contraction induces sand grain rearrangement. This perfectly-plastic sand grain rearrangement produces uplift against gravity ($W_{grav}$) rather than further increases to off-fault deformation ($W_{int}$). Crustal accretionary wedges may deform somewhere between these two end-member rheologies. Sediment aggregates within crustal accretionary wedges are neither ideally linear-elastic nor elastic-perfectly plastic. While these aggregates cannot easily rearrange in response to internal strain, they can develop small folds, faults and microstructures that reduce internal work, and accommodate vertical thickening and horizontal shortening (e.g., *Morgan & Karig,* 1995).

Recent advanced in field techniques provide insights into the magnitude of $W_{fric}$ during seismic slip. Heat anomalies detected following the Tohoku-Oki earthquake indicate that the earthquake produced 27 MJ/m² of frictional heat energy per fault area, with 19-51 MJ/m² in the 91% confidence interval range (*Fulton et al.*, 2013). The low coseismic friction (0.08) estimated by *Fulton et al.* (2013) suggests that the heat produced by frictional sliding may consume a small portion of the total energy budget. Frictional work may be a smaller contribution to the energy budget in crustal environments than in the physical accretion experiments if fluids, melting, thermal pressurization and other processes decrease the effective friction of crustal faults during slip. Future energy budget analyses should endeavor to estimate the proportion of energy consumed in frictional sliding, and other energy budget components, such as the production of fault damage zones, from field observations (e.g., *Shipton et al.*, 2006). The knowledge of the tectonic environments and conditions under which $W_{fric}$ dominates the energy budget may enable estimates of the total system efficiency, $W_{ext}$, from $W_{fric}$ alone. Because numerical analyses demonstrate that $W_{ext}$ can successfully predict fault development (e.g., *McBeck et al.*, 2016, 2017), the ability to estimate total system efficiency from only $W_{fric}$ may enable robust field predictions of fault development.

### 5.3. Impact of apparatus design on strain partitioning

We expected the opposing apparatus setup to exert only a minimal impact on deformation within the wedge, and the resulting energy budget components, because the ratio of sandpack area in contact with the sidewalls and in contact with the base is <0.1 (*Souloumiac et al.*, 2012). However, this ratio was determined from experiments in which the materials in contact with the base and sidewalls were the same, and so it may not be valid for the experiments with glass beads. If the frictional sliding of the sandpack against the sidewalls has an appreciable influence on thrust fault geometry, then the trace of the thrust faults would curve near the sidewalls. In the top view displacement field, we observe mature thrust fault traces sub-perpendicular to the applied contraction direction across the full width of the sandpack (Fig. 2, Fig. S6). However, when the thrust faults just began to initiate within the center of the sandpack and propagate towards the sidewalls (Fig. 2A), the curvature in the traces of the developing faults was more pronounced than the later mature fault traces (Fig. S6). The resulting difference between the displacement fields viewed through the sidewalls and the displacements that occur within the center of the sandpack suggest that the energy budget components calculated from the sideview displacement fields may differ from those within the center of the sandpack most significantly before the thrust faults develop across the full sandpack width.

The distribution of internal compaction differs in the sets of experiments with differing apparatus configurations. The moving base compacted a larger volume of the sand pack than the moving backwall prior to thrust fault development (Fig. 9). When only the backwall displaced, the horizontal contraction of the top of the wedge only reveals significant contraction within 10-15 mm of the backwall (Fig. 9B), preceding thrust fault development (0.19 mm backwall displacement). When the base displaced, the strain field reveals compaction at further distances from the fixed backwall (>20 mm) (Fig. 9A). The experiment setups differ in the area of sidewall where sand grains move relative to the sidewall. A smaller area of the sandpack slides pass the sidewalls in the moving wall experiment than within the moving base experiment. While the influence of sidewall friction on thrust fault curvature is low, non-zero sidewall friction might produce more diffuse compaction prior to fault formation. These differences in the actively compacting volume influenced the strain done within the compacting volumes visible through the glass sidewalls preceding thrust fault development (Fig. S2). These differences lead to slightly higher estimates of $\delta W_{int}$ in the experiments with moving bases relative to moving backwalls. However, the relative similarity of these $\delta W_{int}$ estimates (<0.01 mJ difference between experiments with moving backwalls and moving bases) suggest that the impact of these differing strain volumes did not significant impact the calculated $\delta W_{in}$.

## 6. Conclusions

This study compared the energetic tradeoffs of frictional slip, uplift and distributed off-fault deformation within physical accretion experiments in order to shed insight on the impact of fault development and detachment strength on deformational processes within accretionary environments. We demonstrate that the new experimental set up used here minimizes the effects of sidewall friction on deformation and the subsequent energy budget estimates. This study is the first effort to quantify four components of the deformational energy budget in accretion physical experiments using force and strain measurements. We use these data to calculate external work, internal work of distributed deformation, work against frictional slip and work of uplift against gravity. Constraining the evolving energy budgets of scaled accretion experiments revealed that thrust fault development increased the overall system efficiency, decreasing the incremental external work. Thrust fault development decreased the incremental external work and incremental internal work, and increased the incremental frictional work and incremental work against gravity. The evolution of gravitational work highlighted the broad uplift of the sandpack that begins prior to slip on localized thrust faults. Variations in the frictional work reveal that lower friction coefficients, as well as lower slip on the backthrust and detachment in the wedges with lower strength detachments suppressed the increase in frictional work arising from thrust fault development in these experiments relative to the higher strength detachment experiments. The imbalance of the cumulative energy budget indicates that additional processes besides those captured here, such as the work of propagation and acoustic energy, may consume a significant portion of the energy budget.

**Acknowledgements**

This work was supported in part by a Geological Society of America Student Research Grant and an International Association of Mathematical Geologists Student Research Grant to JM, and NSF grant EAR-1650368 to MC. The experimental data, including incremental displacement fields and force measurements, are available on the GFZ Repository (*McBeck et al.*, 2018). The authors thank Topical Editor Ylona van Dinther, and reviewers Matthias Rosenau and Arthur Bauville for constructive suggestions that improved the paper.

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

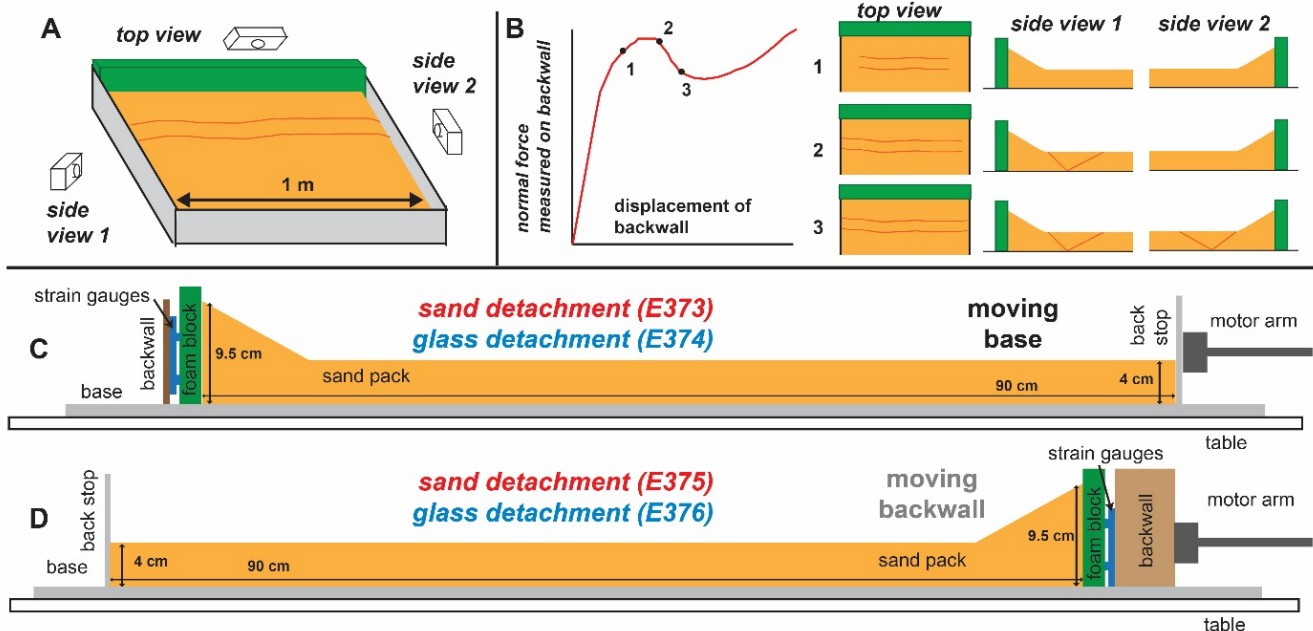

**Figure 1: Schematics of experimental setup, force and faulting evolution. A) Position of cameras relative to sandpack. Backwall and glass**
**sidewalls are in green and gray. B) Characteristic force curve and faulting events revealed through the top and side views. C-D)**
**Experiment apparatus configurations of each experiment. The four experiments differ in the apparatus configuration: C) moving base**
**(E373, E374) or (D) moving backwall (E375, E376), and the inclusion of basal glass bead layer.**

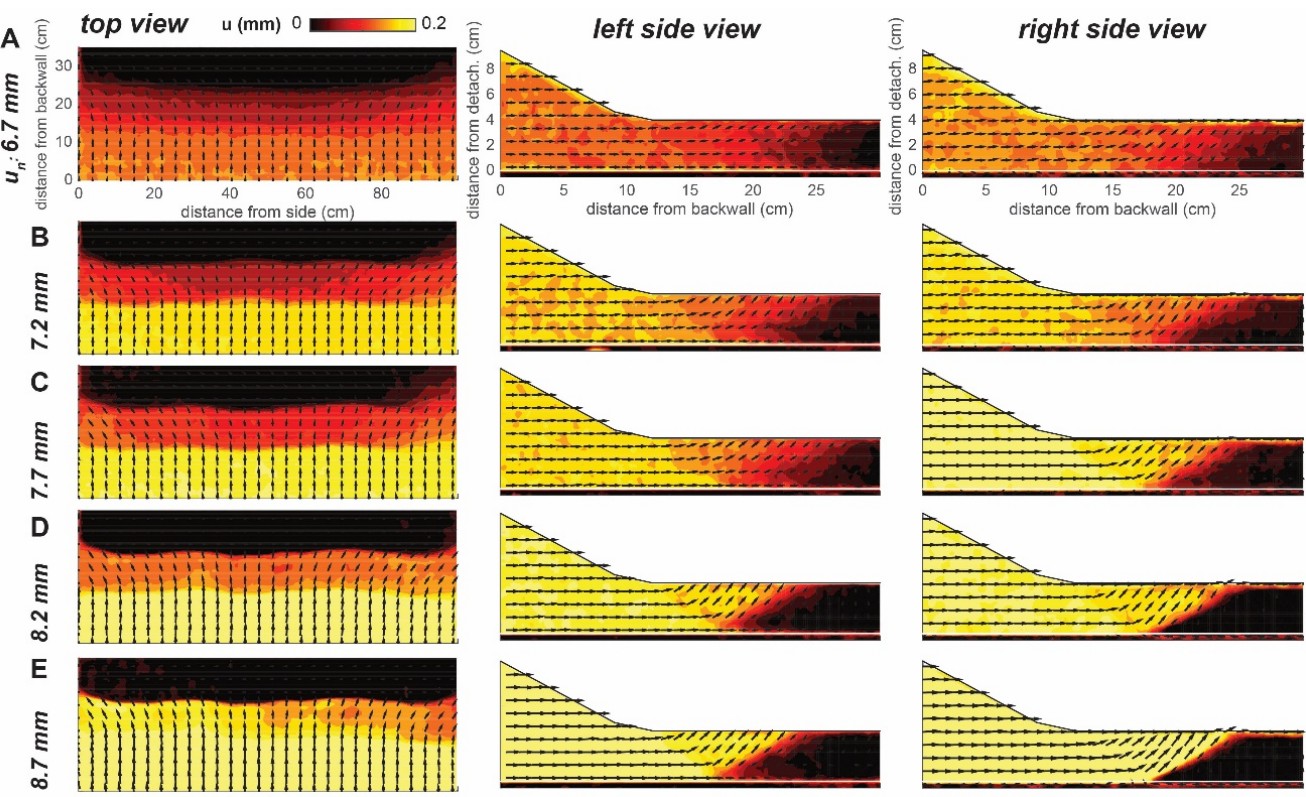

**Figure 2: Incremental displacement fields of top and side views of experiment E374 (glass detachment and moving base) throughout the development of the first thrust pair at 0.5 mm increments of applied backwall displacement, $u_n$. A) Preceding thrust fault development, the net displacement gradually decreases from the backwall toward the opposite wall. B) When sharp gradients in the top view displacement field revealed the development of the first thrust pair within the center of the box, the side view displacement fields showed more diffuse zones of shear. C-E) With continued backwall displacement, the gradients in the top and side view displacement fields progressively sharpened as faults localize. The top view displacement fields reveal that the thrusts began to localize within the center of the sandpack, and then extended toward the sidewalls. The side view displacement fields indicate that fault localization began earlier on the right side than the left side in this experiment (C). The forethrust remained more diffuse on the left side view than the right side view in the last increment shown here (E).**

**Figure 3: Backwall normal force curves with faulting events. A)**
**Force curves throughout complete experiments. The development**
**of thrust pairs produced larger force drops than the development**
**of the singular thrusts. The singular thrusts produced smaller**
**force drops in the experiments with glass detachments than in the**
**experiments with the sand detachments. B-E) Force drops**
**associated with the first episode of faulting in each experiment.**
**Vertical lines indicate when we first observed evidence of localized**
**faulting on the top (black), left side (yellow), and right side (light**
**blue) of the sandpack. Triangles indicate local maximum (peak,**
**upward triangle) and local minimum (trough, side triangle) of**
**first force drop.**

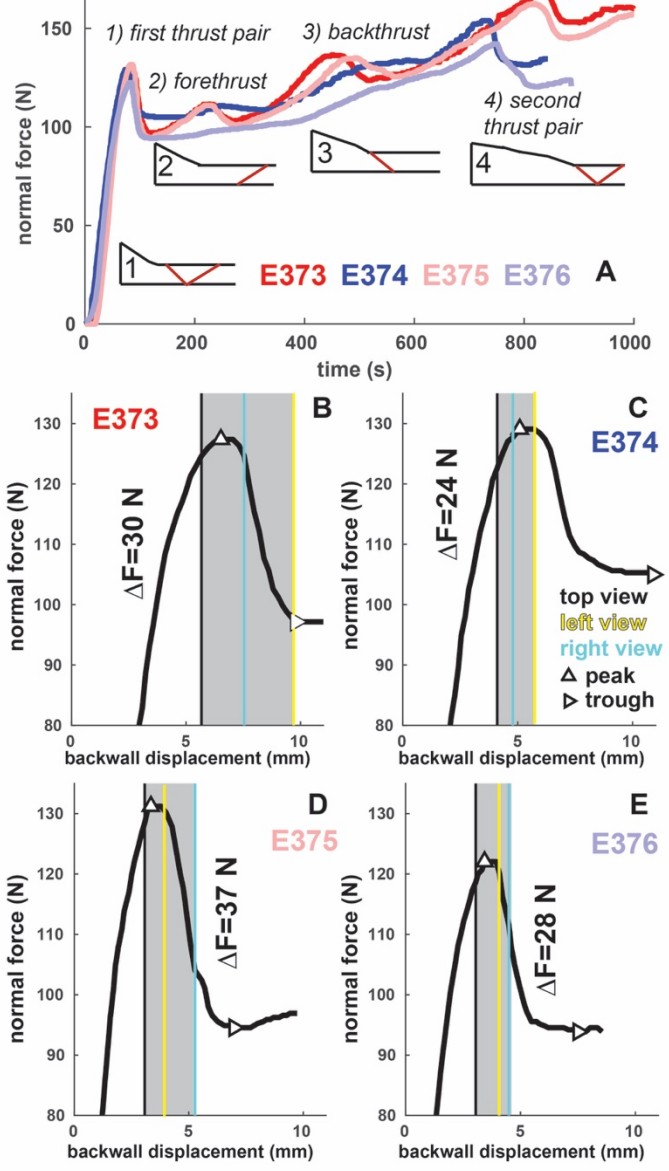

**Figure 4: Incremental work budget components throughout the development of first thrust pair for experiments with the sand detachment (left column), and glass detachment (right column). A) Incremental external work, δW$_{ext}$. B) Incremental internal work, δW$_{int}$. C) Incremental frictional work, δW$_{fric}$. D) Incremental gravitational work, δW$_{grav}$. Vertical dashed and solid lines show stages before and after fault development, respectively. We used the displacement fields observed on the left (upward facing triangles and squares) and right (right facing triangles and circles) sides of the box to calculate δW$_{int}$, δW$_{fric}$ and δW$_{grav}$. Shaded regions are bounded by curves that are 5-point (for δW$_{ext}$) or 20-point (for the other components) median filters through the data obtained from each work budget component. We use a smaller filter window for δW$_{ext}$ in order to capture fluctuations produced by variations in backwall force, which are lost with a 20-point median filter.**

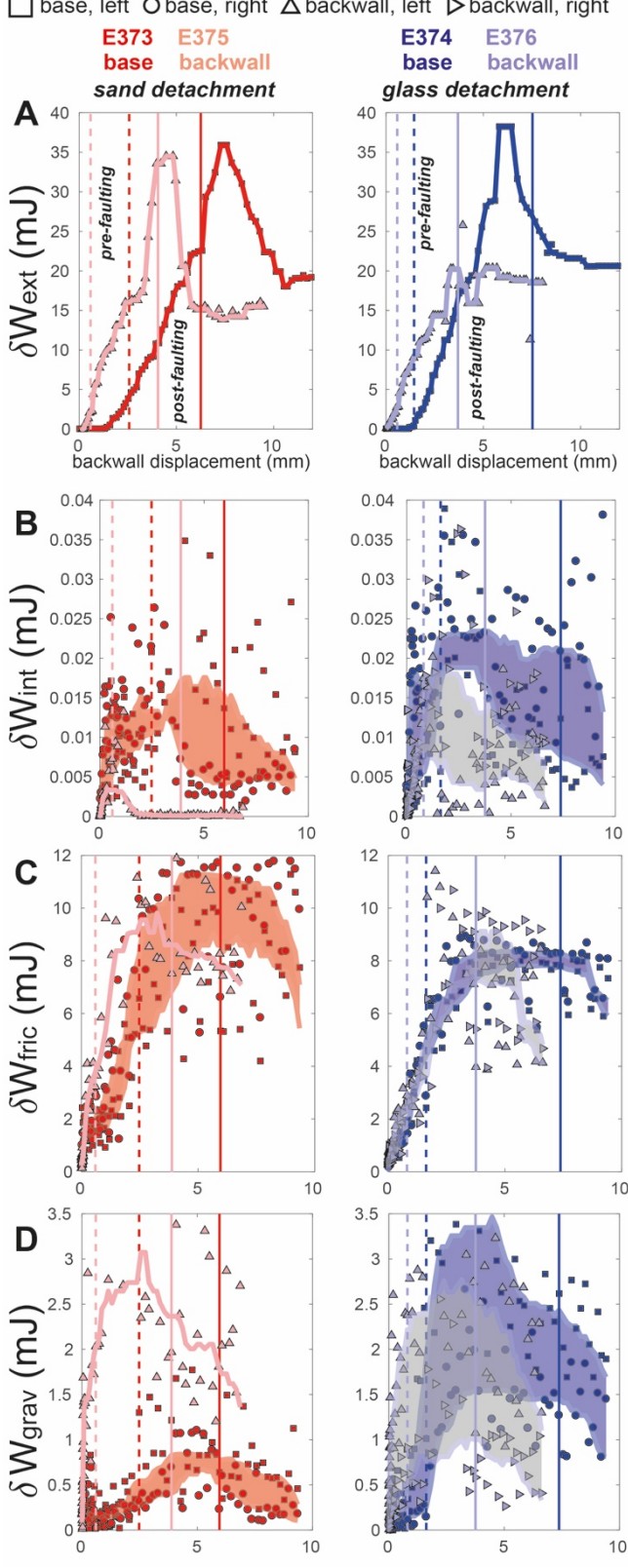

**Figure 5: Vertical displacement fields of moving backwall experiment excluding glass bead layer (E375) preceding (A) and following (B) thrust fault development. Preceding thrust development, a broad pattern of uplift forms in the region where the faults ultimately localize. Following thrust development, uplift localizes above discrete thrust faults. As the uplift pattern transitions from gradational to sharply bounded by faults, the overall $\delta W_{grav}$ increased (Fig. 4).**

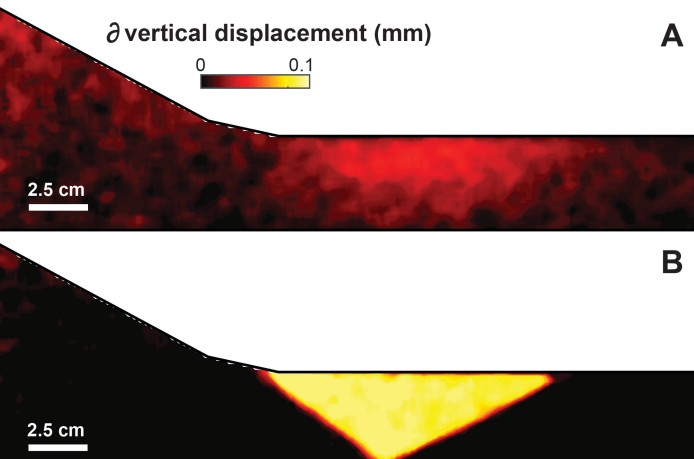

**Figure 6: Shear strain of incremental displacement fields of experiments excluding (A) and including (B) glass bead layer following faulting. The shear strain fields reveal that the inclusion of a glass bead layer suppresses slip on the thrusts. C) Photo of side of experiment E376 following faulting. The entrainment of glass beads along the forethrust promoted slip on the forethrust and suppressed the localization of the backthrust, perhaps by producing a smoother transition zone near the base of the forethrust. This suppression of slip on the thrusts reduced the change in $\delta W_{fric}$ produced by thrust faulting in experiment E374 (glass detachment) relative to that change in experiment E375 (sand detachment).**

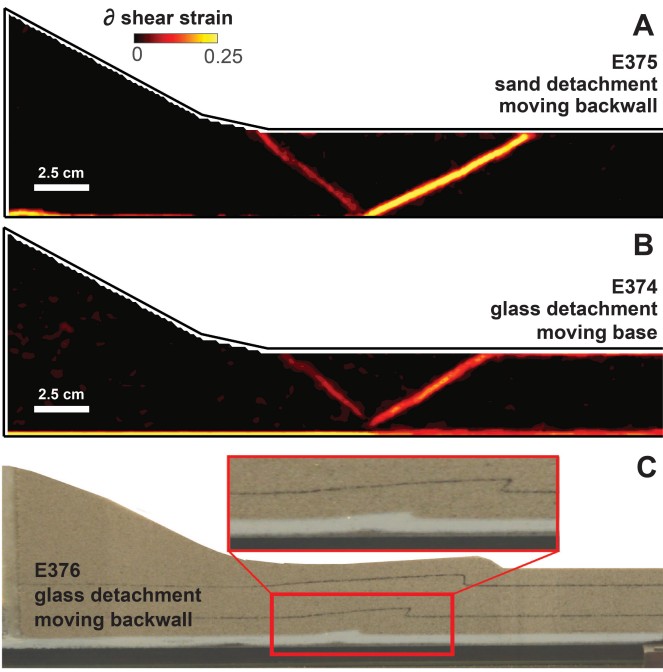

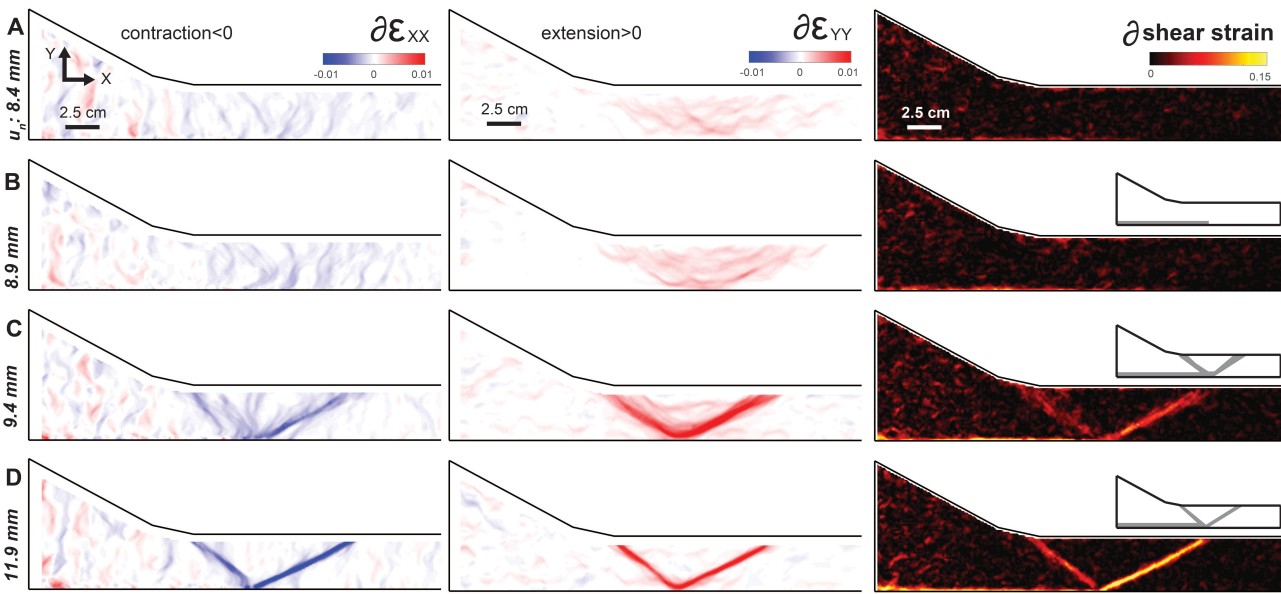

Figure 7: Evolution of strain as the thrust faults develop in experiment E375. Gray regions in sketches in right column indicate portion of strain field that contributes to $\delta W_{fric}$, and the remaining white area contributes to $\delta W_{int}$. A) Preceding thrust faulting, the wedge had distributed high horizontal and vertical normal strains. B) With continued backwall displacement, horizontal contraction and vertical extension shifted toward the region where the thrusts ultimately develop. C) In the incipient stages of thrust development, horizontal contraction and vertical extension formed broad bands of high strain a few centimeters thick, and shear strain began to localize along both thrusts. At this stage, and in subsequent increments, the forethrust produced higher shear strain than the backthrust. D) The zones of high horizontal contraction, vertical extension and shear strain localized into discrete zones of higher incremental strains along the thrusts. The progressively localizing normal and shear strains produced the gradually increasing and decreasing $\delta W_{int}$ coincident with fault development. Preceding fault development, the distributed strains produced maxima values of $\delta W_{int}$. As strain localizes onto faults, $\delta W_{int}$ decreased to near zero, while $\delta W_{fric}$ increased.

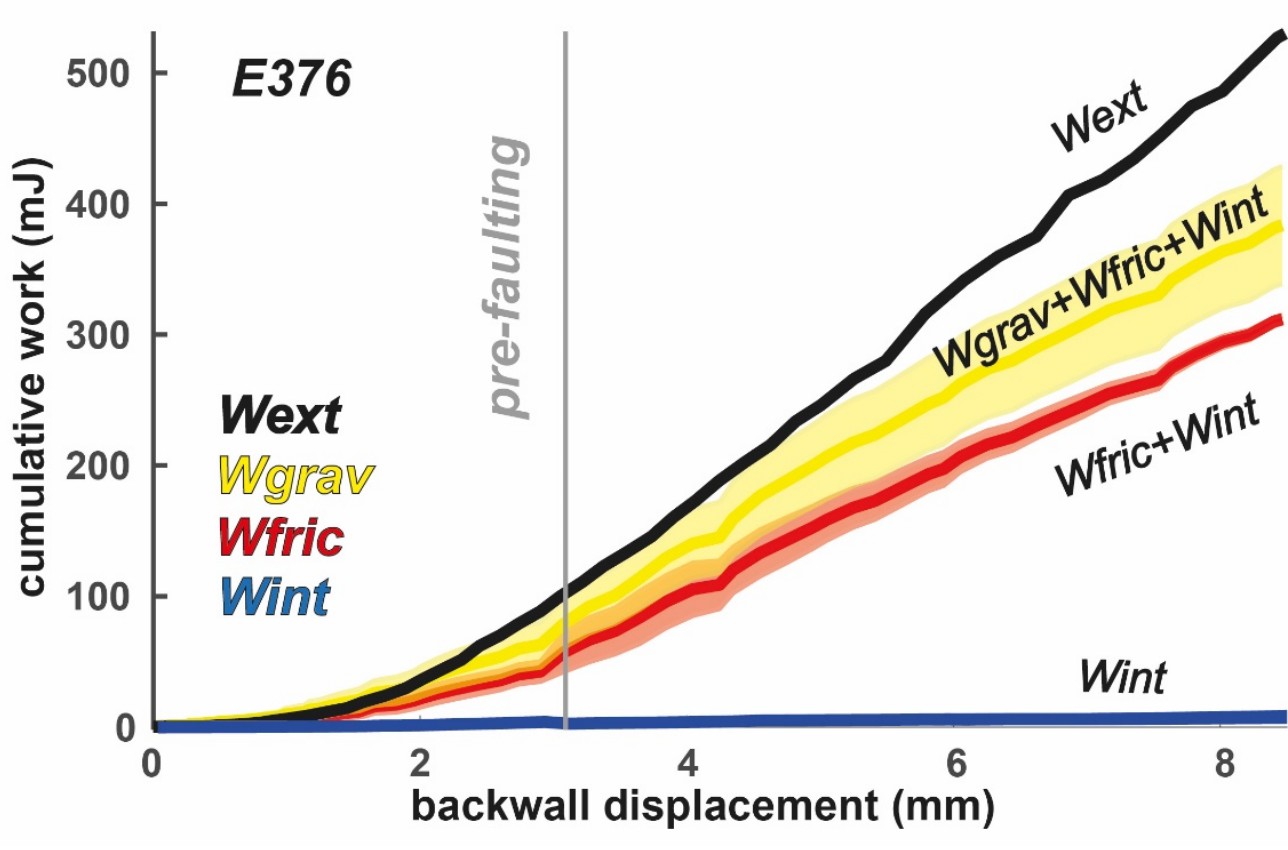

**Figure 8: Total accumulation of work done throughout experiment E376. Upper black line shows total W$_{ext}$. Thicker lines show accumulation of work components calculated from average of the left and right displacement fields. Upper and lower edges of shaded regions show work accumulation calculated from left and right side views, respectively.**

**Figure 9: Normal strain perpendicular to the backwall from top views of experiment A) E374 (moving base) and B) E376 (moving backwall). Contraction is negative. Y-axis is perpendicular to the backwall. Red numbers show the cumulative backwall displacement. The motor is located at the top in the moving base (A) and at the bottom in the moving backwall (B) experiment. Preceding thrust fault development, the extent of the compacting region differed depending on the apparatus configuration. This difference produces changes in the effective stiffness estimates, and internal work estimates.**

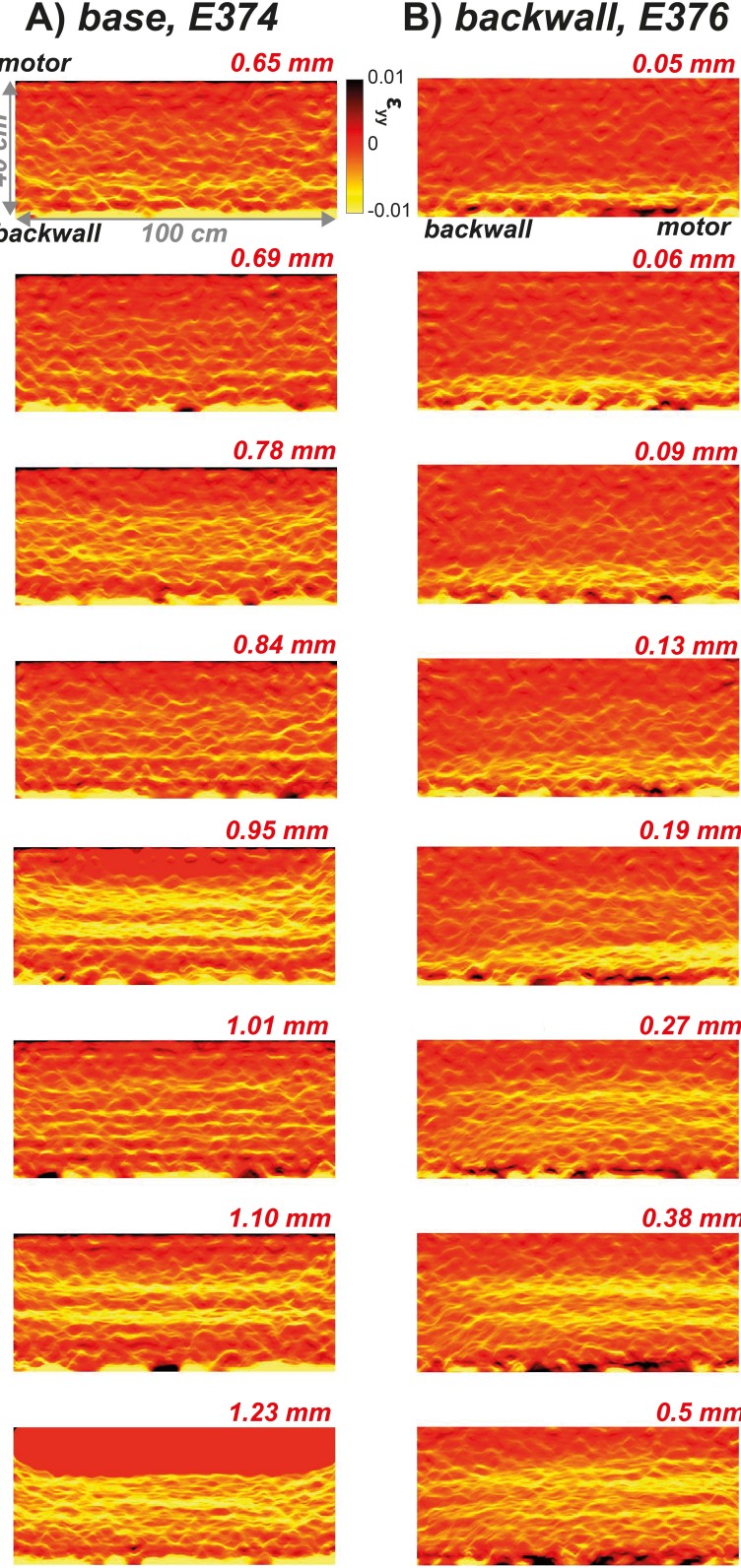