# Peer review of "The influence of detachment strength on the evolving deformational energy budget of physical accretionary prisms"

_Solid Earth, 2018_

## Referee Comment (RC1) · A. Bauville (Referee) · 12 Oct 2018

General comments:

This article presents a set of laboratory compressive sandbox experiments in association with a detailed analysis and discussion of the mechanical work of the sand pack. During the experiment, the authors monitored forces on the back wall using pressure plates and particle displacement on the side using digital image correlation (DIC). The paper is innovative from the technical point of view. Indeed, using back wall force measurements and DIC, and calling on mechanically motivated assumptions, the authors managed to estimate individual components of the mechanical work in a laboratory

experiment. The conclusions confirm the theoretical predictions that were developed in previous publications by the authors. The paper is well organized, the language is good, and the figures are clear. Figures representing graphs are well used (Fig. 3, 4, 8) and are useful to the discussion. Some of the figures showing the experiment and DIC analysis are used more anecdotally (Fig. 2, 5). The literature review is mainly focused on the previous work of the authors and could benefit from being a bit broader, especially regarding older works. In conclusion, this paper is an interesting contribution to the mechanical work of compressive tectonic systems. The focus of the paper is adapted to the journal Solid Earth. I support the publication of this manuscript after some minor corrections.

Arthur Bauville

Specific comments:

As stated above, the literature review is a bit authors-centric. I would like to suggest some readings to broaden a bit the introduction. I listed them at the end of this review.

Fig. 2 and 5 are just used in passing, as a support to Fig. 3 and 4. They could be used a bit better. For example Fig. 2 could be the object of a small paragraph that discusses the dynamics of the experiment and the main events happening before to dive into Fig. 3. In addition, the pre-faulting and post-faulting phases are important events in Fig. 4. The limit pre-post faulting could be indicate on Fig. 2 as well.

Methods A value for the basal friction in the glass and sand detachment experiments should be given here already (it's only given much later in the text).

5.59: "We calculated the incremental shear strain fields of each of the views of the sand pack using the curl of the incremental displacement field. " The curl of the incremental displacement field must be the vorticity*char_time, i.e., the rotational part of deformation instead of strain. In 2D the vorticity is a scalar while strain is a tensor. Please explain better what the quantity delta shear strain corresponds to: invariant of the incremental strain tensor? abs(vorticity)? 6.96. Following the previous comments, here, what do you call the shear strain field? the shear strain tensor field of the displacement curl field?

Discussion 12.62-65. Here you argue that using lithostatic pressure as the traction on the fault may lead to underestimating Wfric. It would be nice to have an estimate of how much that is. For example, an upper estimate can be given by considering that sigma1 is horizontal and deviatoric stresses are such that yielding occurs. Doing some back of the enveloppe trigonometry on the Mohr-Coulomb diagram I get: Sigma_n = Sigma_mean- (Pl*sin(phi)*tan(phi))/sin(1-phi) where Pl is the lithostatic pressure and phi is the friction angle (you might want to double check my math). Using a friction angle of 30$^\circ$ it simplifies to Sigma_n = 2*Pl - (Pl*sin(phi)*tan(phi))/sin(1-phi).

That yields Sigma_n/Pl = 1.35; and at the end of the experiment Wext/(Wgrav+Wfric+Wint)=1.35. That's pretty nice :)

Sigma1 is probably not horizontal which gives you some latitude to put Wprop and Wseism.

Also, the discussion would be a bit more general if you expressed the deficit in percentage in addition to giving the values in mJ.

Technical corrections

2.42: Ritter et al., 2018 -> ref missing 3.72: "In these numerical accretion simulations, all of the work components increase during underthrusting. The development of the new forethrust increases Wint, but decreases Wfric by a greater degree, which correspondingly decreases Wext (Del Castello and Cooke, 2007)." The second sentence seems to contradict the first one.

4.05 Missing citation for Dotare et al., 2016. Possibly miscited. Do you mean this paper?: Yasuhiro Yamada, Tatsuya Dotare, Juergen Adam, Takane Hori, Hide Sakaguchi. Initiation process of a thrust fault revealed by analog experiments. Geophysical Research Abstracts, 2016, 18 4.25. Reverse the order of citations 9.60. Fig. 3 is cited before Fig. 2. The paragraph can be easily rearranged to avoid that.

9.69. Earlier work could be cited (e.g. Vermeer, 1990)

9.88. Unneeded reference to Fig. 5B 9.93. There is no Fig. 5C 9.94 primary->primarily 10.5. Here you can refer to Fig. 5

Minor comments on figures:

Fig. 3 Panel A has a lot of curves. The curve from exp E374 is hidden behind the others which is a bit unfortunate. Since E373 and E374 are the most important, the drawing order could be changed (i.e. as in the legend). Using thinner lines for E375, E376 could also improe readability Panel B etc-E the light blue curve is hard to distinguish from the grey background

Suggested readings

Vermeer P. (1990). – The orientation of shear bands. – Géotechnique, 40, 223-234.

Le Pourhiet, Laetitia. "Strain localization due to structural softening during pressure sensitive rate independent yielding."ÂăBulletin de la Société Géologique de FranceÂă184.4-5 (2013): 357-371.

Dahlen, F. A. "Mechanical energy budget of a fold-and-thrust belt."ÂăNatureÂă331.6154 (1988): 335.

Gutscher, Marc‐André, et al. "Episodic imbricate thrusting and underthrusting: Analog experiments and mechanical analysis applied to the Alaskan accretionary wedge."ÂăJournal of Geophysical Research: Solid EarthÂă103.B5 (1998): 10161-10176.

---

## Referee Comment (RC2) · M. Rosenau (Referee) · 15 Oct 2018

Review of McBeck et al. "The influence of detachment strength. . ."

The paper describes sandbox experiments designed to shed light on the work done in deforming accretionary wedges. It continues a series of papers by the group in considering additional terms of the work budget. A secondary issue is the comparison between two endmember setups of the archetypical sandbox often referred to as "push" vs. "pull". It uses state of the art strain monitoring and force sensing techniques to derive at a more complete work budget formulation.

[Figure]

General comments:

I think this is a landmark paper for modern analysis of a classical experiment and a big step towards a complete work budget of sandbox experiments. The latter is of prime importance when arguing about the dynamic similarity with natural accretionary wedges. Also, giving the increasing resolution of experimental observations requires a re-assessment of the similarity in energetic terms. Only then, new interpretations and implications for nature are possible. I therefore think this paper is a timely and important contribution to be considered for publication after some minor issues are solved as suggested below.

One point that remains unclear to me is the role of Wint (internal deformation). You describe it as elastic strain energy, but is it recoverable? From my experience, I would argue that distributed plastic deformation (compaction) takes up quite a substantial amount (few percent) of the external work applied and should be considered a part of Wint? Could it be useful to add or split off another "damage" term describing the plastic internal work done? Also, the rigidity assumption (1 MPa) seems to me at the lower end (like for a low density, unconfined grain network rather than a well compacted, 5/6-sides confined sand pack) and increasing it could close the gap seen in the work budget. See my comments below on the respective section.

Since the paper has the potential to be an important contribution to the sandbox community, I would suggest adding a paragraph in the discussion where a comparison with natural wedges, and work done within them, is tried. I think it could be useful to do this to get an idea of how similar experiments are to nature in energetic terms and consequently what new inferences could be drawn from sandbox experiments for the prototype. Given the increasing amount of quantitative observations, the limits of interpretation should be well respected and I think this paper can help a lot in defining those limits.

A minor point is referencing the work of Malte Ritter. In the paper you cite Ritter et

al. (2018) but it is not in the reference list, so I cannot judge which of his papers you mean. Since he published two papers in 2018 and one in 2016 (all Ritter et al.), that all fit neatly into this topic, I tried to sort out things below and make suggestions how to include his work here. I think his papers can serve to support your findings nicely.

Finally, I suspect you will publish data using GFZ Data Services in the framework of EPOS. Please contact GFZ data services soon enough to allow inclusion of the reference in this paper and register them as "assets" for this article.   Detailed comments:

Page 2 line 42: Ritter et al. (2018): you probably refer to Ritter et al. (2018a, see below). It needs to be added to the Reference list.

Page 4 Line 7 f: The "sandbox rheometry" models by Ritter et al. (2018b) give additional insights into the localization process in wedge experiments. They clearly observe an increase in total work done which is correlated with the onset of diffuse deformation prior to localization. This confirms your hypothesis "Prior to faulting distributed internal strain (Wint) may accommodate a larger percentage of the overall work budget than after thrust fault development,. . ." and may serve as reference here.

Page 5 Line 30 ff: What is the size of the glassbeads used?

Page 5 line 53 ff: There could be some more, basic information about the imaging setup used (SLR? How much MPx), treatment of distoration, calibration procedure, final resolution of incremental vectorfield/strain field (now in chapter 3.6), imaging frequency with respect to backwall movement (now on page 6 line 75f), software used for DIC. This is very useful information not only for evaluating the quality of your DIC analysis but will appreciated also by those people setting up new labs and interested in the way to do it.

Page 5 line 57: You refer here to the PHD thesis of Silvan Hoth for DIC. Actually, Adam et al. (2005) is the more proper reference for application of DIC/PIV to sandbox models.

Page 6 line 74: Why should there be a non-steady state backwall motion. Is this due to the motor, sticky parts of the compliancy of the force sensor-armed backwall? Could this be quantified? I ask because in the end you compare the force readings (at regular temporal intervals) with strain increments (not necessarily at the same regular temporal intervals).

Page 7 line 7ff: It is not entirely clear to me how the distributed strain (diffuse compaction) is related to elastic strain (which should be too small to be observed) given the rigidity of sandpacks? In Figure S2 you use quasi-linear segment of the strain hardening curve – is this really the elastic loading path? The values around 1 MPa appear, as you describe in the Appendix, too low. From every day life experience 1 MPa is what foam has as a Young modulus. What would happen if you consider 100 MPa in your calculation, do the numbers become unrealistic, or may this fill the gap in the work budget described in Ch. 5.1? In general it is not quite clear to me how you calculate Wint (you refer to Cooke and Madden but it would be good to recall it here with a formula and/or figure). When I understand correctly you use the curves such as in fig S2 to derive the stress/strain relationship (elastic modulus) and calculate strain based on the assumption that backwall push is transmitted all over to the opposite side of the experiment (i.e. ratio of backwall displacement and experiment length). If so, I would argue that actual strain is underestimated as experiments with force sensors on both sides (Maillot et al.) showed that force is transmitted only at later stage of such an experiment, when the decollement is closer to the opposite side.

Page 8 line 41ff: The kinematic compatibility assessment is highly appreciated but it may be better placed into the appendix because of the technical character. Probably it is OK if you put the conclusion ("Assessment of the accuracy and precision of the method results in a resolution of incremental vector field of about XX px / YY mm") in chapter 3.2.

Page 8 line 55 ff: To better understand the general model evolution I suggest to prepend a short qualitative description of the evolution of the experiments (sequence of faulting,

how many thrust in total,. . .) before starting the detailed description. See also comment on Fig3 a below.

Page 9 line 70: Ritter et al. (2016) /not (2018) is the actual reference for the weakening behavior of sand and glassbeads faults (also consistent btw with lower absolute stress drops in glassbeads compared to sand).

Page 10 line 22f and 26f: There seems to be redundancy here!?

Page 11 line 31: Fig. 4C not 5C

Page 12 line 74: Why is Wgrowth not considered here, is it impossible to constrain from the setup?

Figure 3

A: - text: "glass bead detachment"; - it's a bit busy with the combination of 4 setups. Either a different colorcode and bigger, or two plots? - I don't quite understand: Are the peaks labelled fore- and backthrust secondary peaks following the "first thrust pair" peak. In other words: Do those fore-and backthrusts represent the first thrust pair? From the text it appears that these are new thrusts forming after the first pair? To clarify in gnereal, I suggest to add a sequence of images showing each stage of new thrust formation covering the full experimental run (the 1000 seconds), probably in appendix? Or a movie of each experiment? And a short qualitative description of the evolution of the experiments in chapter 4.

B-E: - The light vs dark blue is difficult to see. I suggest to use more different colours.

Figure 4:

- Also quite busy the figures and the different markers and their assignment to experiments and view are not easy to capture.z - Maybe use two transparent bands instead 4 lines to indicate the phase of localization. - Maybe display experiment in individual panels.

Page 14 line 34f: I suspect you will publish your data with GFZ Data Services in the framework of EPOS. I so, the sentence should be "Data. . .. Have been uploaded to the GFZ Repository published open access in McBeck et al. (2018)".

References:

Adam, J., J.L. Urai, B. Wieneke, O. Oncken, K. Pfeiffer, N. Kukowski, J. Lohrmann, S. Hoth, W. van der Zee, J. Schmatz, (2005), Shear localisation and strain distribution during tectonic faulting - New insights from granular-flow experiments and high-resolution optical image correlation techniques, Journal of Structural Geology, 27(2), 183-301,doi:10.1016/j.jsg.2004.08.008.

Ritter, M., K. Leever, M. Rosenau, and O. Oncken (2016): Scaling the Sand Box - Mechanical (Dis-) Similarities of Granular Materials and Brittle Rock, J. Geophys. Res - Solid Earth, doi.: 10.1002/2016JB012915.

Ritter, M.C., M. Rosenau, O. Oncken (2018a): Growing Faults in the Lab: Insights into the Scale Dependence of the Fault Zone Evolution Process, Tectonics, doi: 10.1002/2017TC004787

Ritter, M.C., T. Santimano, M. Rosenau, K. Leever, O. Oncken (2018b): Sand-box rheometry: Co-evolution of stress and strain in Riedel– and Critical Wedge– experiments, Tectonophysics, Volume 722, 2 January 2018, Pages 400-409, ISSN 0040-1951, doi: 10.1016/j.tecto.2017.11.018.

——————end of review——————

---

## Author Comment (AC1) · 19 Nov 2018

*Response to Revisions*

**Dear Editor van Dinther, Dr. Bauville and Dr. Rosenau,**

**Thank you for these careful revisions. We have made several changes following your constructive comments. The most significant of these changes includes a new supplementary figure (Fig. S5) that shows how frictional work changes when we assume varying ratios of principal stresses (in which the tectonic normal compression exceeds the lithostatic), and how internal work changes when we use different elastic moduli to convert strains to stresses.**

**We respond to individual comments point-by-point below with bolded text. We number our responses for clarity.**

**Sincerely,**
**Jessica McBeck**

*Comments from Referee:*

**A. Bauville (Referee)**

abauville@jamstec.go.jp

General comments:

This article presents a set of laboratory compressive sandbox experiments in association with a detailed analysis and discussion of the mechanical work of the sand pack. During the experiment, the authors monitored forces on the back wall using pressure plates and particle displacement on the side using digital image correlation (DIC). The paper is innovative from the technical point of view. Indeed, using back wall force measurements and DIC, and calling on mechanically motivated assumptions, the authors managed to estimate individual components of the mechanical work in a laboratory experiment. The conclusions confirm the theoretical predictions that were developed in previous publications by the authors. The paper is well organized, the language is good, and the figures are clear. Figures representing graphs are well used (Fig. 3, 4, 8) and are useful to the discussion. Some of the figures showing the experiment and DIC analysis are used more anecdotally (Fig. 2, 5). The literature review is mainly focused on the previous work of the authors and could benefit from being a bit broader, especially regarding older works. In conclusion, this paper is an interesting contribution to the mechanical work of compressive tectonic systems. The focus of the paper is adapted to the journal Solid Earth. I support the publication of this manuscript after some minor corrections.

Arthur Bauville

Specific comments:

As stated above, the literature review is a bit authors-centric. I would like to suggest some readings to broaden a bit the introduction. I listed them at the end of this review.

1) **This is a good point. We now have included additional references throughout the manuscript.**

Fig. 2 and 5 are just used in passing, as a support to Fig. 3 and 4. They could be used a bit better. For example Fig. 2 could be the object of a small paragraph that discusses the dynamics of the experiment and the main events happening before to dive into Fig. 3. In addition, the pre-faulting and post-faulting phases are important events in Fig. 4. The limit pre-post faulting could be indicate on Fig. 2 as well.

2) **We agree that Fig. 2 and Fig. 5 deserve more attention. We now have discussed them in more detail (lines 278-286, lines 349-356).**

Methods A value for the basal friction in the glass and sand detachment experiments should be given here already (it's only given much later in the text).

3) **This important detail is in the Methods section 3.4 (line 223).**

5.59: "We calculated the incremental shear strain fields of each of the views of the sand pack using the curl of the incremental displacement field. " The curl of the incremental displacement field must be the vorticity*char_time, i.e., the rotational part of deformation instead of strain. In 2D the vorticity is a scalar while strain is a tensor. Please explain better what the quantity delta shear strain corresponds to: invariant of the incremental strain tensor? abs(vorticity)? 6.96. Following the previous comments, here, what do you call the shear strain field? the shear strain tensor field of the displacement curl field?

4) **For small angles of rotation, the tangent of that angle is equal to the angle. So, we can calculate the incremental shear strain as the absolute value of the curl of the incremental displacement field. We now include this specification in the methods (lines 166-170).**

Discussion 12.62-65. Here you argue that using lithostatic pressure as the traction on the fault may lead to underestimating Wfric. It would be nice to have an estimate of how much that is. For example, an upper estimate can be given by considering that sigma1 is horizontal and deviatoric stresses are such that yielding occurs. Doing some back of the enveloppe trigonometry on the Mohr-Coulomb diagram I get: Sigma_n = Sigma_mean-(Pl*sin(phi)*tan(phi))/sin(1-phi) where Pl is the lithostatic pressure and phi is the friction angle (you might want to double check my math). Using a friction angle of $30°$ it simplifies to Sigma_n = 2*Pl - (Pl*sin(phi)*tan(phi))/sin(1-phi).

That yields Sigma_n/Pl = 1.35; and at the end of the experiment Wext/(Wgrav+Wfric+Wint)=1.35. That's pretty nice :)

Sigma1 is probably not horizontal which gives you some latitude to put Wprop and Wseism.

5) **We agree that estimating the potential error induced by assuming lithostatic normal compression will benefit this analysis. We have now added a new figure (Fig. S5) showing the impact of using higher estimates of normal compression on the frictional work, and discuss this more thoroughly in the text (lines 395-399).**

Also, the discussion would be a bit more general if you expressed the deficit in percentage in addition to giving the values in mJ.

**6) We have now added this percentage (line 384).**

Technical corrections

2.42: Ritter et al., 2018 -> ref missing

**7) We have added this reference.**

3.72: "In these numerical accretion simulations, all of the work components increase during underthrusting. The development of the new forethrust increases Wint, but decreases Wfric by a greater degree, which correspondingly decreases Wext (Del Castello and Cooke, 2007)." The second sentence seems to contradict the first one.

**8) We have corrected this mistake (lines 73-75).**

4.05 Missing citation for Dotare et al., 2016. Possibly miscited. Do you mean this paper?: Yasuhiro Yamada, Tatsuya Dotare, Juergen Adam, Takane Hori, Hide Sakaguchi. Initiation process of a thrust fault revealed by analog experiments. Geophysical Research Abstracts, 2016, 18

**9) We have added the correct reference (Dotare et al., 2016) to the list.**

4.25. Reverse the order of citations

**10) We have corrected this (line 134).**

9.60. Fig. 3 is cited before Fig. 2. The paragraph can be easily rearranged to avoid that.

**11) We have corrected this with a new paragraph (lines 280-286)**

9.69. Earlier work could be cited (e.g. Vermeer, 1990)

**12) We have added this reference (line 297).**

9.88. Unneeded reference to Fig. 5B 9.93. There is no Fig. 5C 9.94 primary->primarily 10.5. Here you can refer to Fig. 5

**13) We have corrected these mistakes (line 322, lines 349-354).**

Minor comments on figures:

Fig. 3 Panel A has a lot of curves. The curve from exp E374 is hidden behind the others which is a bit unfortunate. Since E373 and E374 are the most important, the drawing order could be changed (i.e. as in the legend). Using thinner lines for E375, E376 could also improe readability Panel B etc-E the light blue curve is hard to distinguish from the grey background

**14) We have modified this figure to improve clarity.**

Suggested readings

Vermeer P. (1990). – The orientation of shear bands. – Géotechnique, 40, 223-234.

Le Pourhiet, Laetitia. "Strain localization due to structural softening during pres- sure sensitive rate independent yielding."Âa˘Bulletin de la Société Géologique de FranceÂa˘184.4-5 (2013): 357-371.

Dahlen, F. A. "Mechanical energy budget of a fold-and-thrust belt."Âa˘NatureÂa˘331.6154 (1988): 335.

Gutscher, MarcâAˇRˇAndré, et al. "Episodic imbricate thrusting and underthrusting: Analog experiments and mechanical analysis applied to the Alaskan accretionary wedge."Âa˘Journal of Geophysical Research: Solid EarthÂa˘103.B5 (1998): 10161- 10176.

**15) We have now added additional citations to previous work throughout the manuscript.**

---

## Author Comment (AC2) · 19 Nov 2018

*Response to Revisions*

**Dear Editor van Dinther, Dr. Bauville and Dr. Rosenau,**

**Thank you for these careful revisions. We have made several changes following your constructive comments. The most significant of these changes includes a new supplementary figure (Fig. S5) that shows how frictional work changes when we assume varying ratios of principal stresses (in which the tectonic normal compression exceeds the lithostatic), and how internal work changes when we use different elastic moduli to convert strains to stresses.**

**We respond to individual comments point-by-point below with bolded text. We number our responses for clarity, continuing from the numbering scheme of the revisions of Dr. Bauville.**

**Sincerely,**
**Jessica McBeck**

*Comments from Referee:*

**M. Rosenau (Referee)**

rosen@gfz-potsdam.de

Review of McBeck et al. "The influence of detachment strength. . ."

The paper describes sandbox experiments designed to shed light on the work done in deforming accretionary wedges. It continues a series of papers by the group in considering additional terms of the work budget. A secondary issue is the comparison between two endmember setups of the archetypical sandbox often referred to as "push" vs. "pull". It uses state of the art strain monitoring and force sensing techniques to derive at a more complete work budget formulation.

General comments:

I think this is a landmark paper for modern analysis of a classical experiment and a big step towards a complete work budget of sandbox experiments. The latter is of prime importance when arguing about the dynamic similarity with natural accretionary wedges. Also, giving the increasing resolution of experimental observations requires a re-assessment of the similarity in energetic terms. Only then, new interpretations and implications for nature are possible. I therefore think this paper is a timely and important contribution to be considered for publication after some minor issues are solved as suggested below.

One point that remains unclear to me is the role of Wint (internal deformation). You describe it as elastic strain energy, but is it recoverable? From my experience, I would argue that distributed plastic deformation (compaction) takes up quite a substantial amount (few percent) of the external work applied and should be considered a part of Wint? Could it be useful to add or split off another "damage" term describing the plastic internal work done? Also, the rigidity assumption (1 MPa) seems to me at the lower end (like for a low density, unconfined

grain network rather than a well compacted, 5/6-sides confined sand pack) and increasing it could close the gap seen in the work budget. See my comments below on the respective section.

**16) Indeed, $W_{int}$, as calculated, would be recoverable. $W_{int}$ is calculated as the volume integral of the strain energy density (SED) in the region outside the faults in this analysis, and in previous work (e.g., *Cooke and Madden*, 2014). Some portion of the stored strain within the sandpack between the faults may be recoverable upon loading and unloading; such recoverability accounts for the drop in external force upon development of new thrust faults. The off-fault strain measured in the experiments captures the elastic, as well as inelastic and plastic, strain. However, by using elastic constitutive properties to estimate the stresses from the strains, the estimated $W_{int}$ reflects the elastic strain energy density. We have now specified this point in the text (lines 228-230).**

**17) We have now added an additional figure showing how the estimates of $W_{int}$ vary with differing assumed effective elastic moduli (Fig. S5).**

Since the paper has the potential to be an important contribution to the sandbox community, I would suggest adding a paragraph in the discussion where a comparison with natural wedges, and work done within them, is tried. I think it could be useful to do this to get an idea of how similar experiments are to nature in energetic terms and consequently what new inferences could be drawn from sandbox experiments for the prototype. Given the increasing amount of quantitative observations, the limits of interpretation should be well respected and I think this paper can help a lot in defining those limits.

**18) We have added discussion of these important points, including potential differences between the experiments and crustal prisms, and the limits of potential interpretations (new Section 5.2).**

A minor point is referencing the work of Malte Ritter. In the paper you cite Ritter et al. (2018) but it is not in the reference list, so I cannot judge which of his papers you mean. Since he published two papers in 2018 and one in 2016 (all Ritter et al.), that all fit neatly into this topic, I tried to sort out things below and make suggestions how to include his work here. I think his papers can serve to support your findings nicely.

**19) We have now added the appropriate reference, and included additional references to this work.**

Finally, I suspect you will publish data using GFZ Data Services in the framework of EPOS. Please contact GFZ data services soon enough to allow inclusion of the reference in this paper and register them as "assets" for this article.

**20) We now include the doi to the GFZ data repository for this work.**

Detailed comments:

Page 2 line 42: Ritter et al. (2018): you probably refer to Ritter et al. (2018a, see below). It needs to be added to the Reference list.

**See #19 above.**

Page 4 Line 7 f: The "sandbox rheometry" models by Ritter et al. (2018b) give additional insights into the localization process in wedge experiments. They clearly observe an increase in total work done which is correlated with the onset of diffuse deformation prior to localization. This confirms your hypothesis "Prior to faulting distributed internal strain (Wint) may accommodate a larger percentage of the overall work budget than after thrust fault development,. . ." and may serve as reference here.

**21) This is a good point, and we have added it to the paper (line 112-114).**

Page 5 Line 30 ff: What is the size of the glassbeads used?

**22) We have now added this important detail to the paper (line 135).**

Page 5 line 53 ff: There could be some more, basic information about the imaging setup used (SLR? How much MPx), treatment of distoration, calibration procedure, final resolution of incremental vectorfield/strain field (now in chapter 3.6), imaging frequency with respect to backwall movement (now on page 6 line 75f), software used for DIC. This is very useful information not only for evaluating the quality of your DIC analysis but will appreciated also by those people setting up new labs and interested in the way to do it.

**23) We have now added these important steps to the Methods section (lines 171-176).**

Page 5 line 57: You refer here to the PHD thesis of Silvan Hoth for DIC. Actually, Adam et al. (2005) is the more proper reference for application of DIC/PIV to sandbox models.

**24) We have replaced this reference with the more appropriate one.**

Page 6 line 74: Why should there be a non-steady state backwall motion. Is this due to the motor, sticky parts of the compliancy of the force sensor-armed backwall? Could this be quantified? I ask because in the end you compare the force readings (at regular temporal intervals) with strain increments (not necessarily at the same regular temporal intervals).

**25) The non-steady backwall motion seems to arise from the screw motor, and not from the sticky parts of backwall-force system. At the slow motor speed used here, we could observe by eye the slowing and speeding up of the motor. So, these differing rates influence the force measurements to the same degree as the strain measurements (observed through DIC). We now include these details in the text (lines 191-194).**

Page 7 line 7ff: It is not entirely clear to me how the distributed strain (diffuse compaction) is related to elastic strain (which should be too small to be observed) given the rigidity of sandpacks? In Figure S2 you use quasi-linear segment of the strain hardening curve – is this really the elastic loading path? The values around 1 MPa appear, as you describe in the Appendix, too low. From every day life experience 1 MPa is what foam has as a Young modulus. What would happen if you consider 100 MPa in your calculation, do the numbers become unrealistic, or may this fill the gap in the work budget described in Ch. 5.1? In general it is not quite clear to me how you calculate Wint (you refer to Cooke and Madden but it would be good to recall it here with a formula and/or figure). When I understand correctly you use the curves such as in fig S2 to derive the stress/strain relationship (elastic modulus)

and calculate strain based on the assumption that backwall push is transmitted all over to the opposite side of the experiment (i.e. ratio of backwall displacement and experiment length). If so, I would argue that actual strain is underestimated as experiments with force sensors on both sides (Maillot et al.) showed that force is transmitted only at later stage of such an experiment, when the decollement is closer to the opposite side.

**26) See #16-17 above, and the new supplemental figure (Fig. S5). In addition, we have included further discussion of potential errors in the effective stiffness estimate in the supplementary information. The low apparent stiffness of the sandpack reflects the ease of the sand grains to rearrange rather than the stiffness of the sand particles. For this reason, the relatively low value of stiffness does not surprise us. In the supplemental information, we now discuss how using a shorter compaction length (rather than the full length of the sandpack) impacts the calculation of internal work. Using a length that matches the extent of the high contraction region near the backwall (20 cm) reduces the effective stiffness estimate, and consequently reduces the estimate of internal work. This reduction does not change our general conclusion that internal work comprises a small portion of the energy budget.**

Page 8 line 41ff: The kinematic compatibility assessment is highly appreciated but it may be better placed into the appendix because of the technical character. Probably it is OK if you put the conclusion ("Assessment of the accuracy and precision of the method results in a resolution of incremental vector field of about XX px / YY mm") in chapter 3.2.

**27) We feel that this new approach, although technical in character, deserves to be mentioned in the main text because of its innovation. Many researchers may find this to be a beneficial approach to assess to robustness of their DIC results.**

Page 8 line 55 ff: To better understand the general model evolution I suggest to prepend a short qualitative description of the evolution of the experiments (sequence of faulting, how many thrust in total,. . .) before starting the detailed description. See also comment on Fig3 a below.

**28) We have now added further description of the general evolution of the experiments (line 282-286), including schematics in Fig. 3.**

Page 9 line 70: Ritter et al. (2016) /not (2018) is the actual reference for the weakening behavior of sand and glassbeads faults (also consistent btw with lower absolute stress drops in glassbeads compared to sand).

**29) We have now provided the correct reference.**

Page 10 line 22f and 26f: There seems to be redundancy here!? Page 11 line 31: Fig. 4C not 5C

**30) We have made these corrections.**

Page 12 line 74: Why is Wgrowth not considered here, is it impossible to constrain from the setup?

**31) We cannot robustly estimate the change in shear stress along the sliding faults, so we cannot provide confident estimates. In Section 5.1, we discuss how the contribution of W$_{seis}$ and W$_{grow}$ could contribute to the work budget deficit.**

Figure 3

A: - text: "glass bead detachment"; - it's a bit busy with the combination of 4 setups. Either a different colorcode and bigger, or two plots? - I don't quite understand: Are the peaks labelled fore- and backthrust secondary peaks following the "first thrust pair" peak. In other words: Do those fore-and backthrusts represent the first thrust pair? From the text it appears that these are new thrusts forming after the first pair? To clarify in gnereal, I suggest to add a sequence of images showing each stage of new thrust formation covering the full experimental run (the 1000 seconds), probably in appendix? Or a movie of each experiment? And a short qualitative description of the evolution of the experiments in chapter 4.

B-E: - The light vs dark blue is difficult to see. I suggest to use more different colours.

**32) We have now improved the layout and coloring of this figure, including schematic images.**

Figure 4:

- Also quite busy the figures and the different markers and their assignment to experiments and view are not easy to capture.z - Maybe use two transparent bands instead 4 lines to indicate the phase of localization. - Maybe display experiment in individual panels.

**33) We agree that this figure can be confusing at first. However, using two transparent bands increases the complexity because these bands overlap on the plots, producing apparently 3 bars. In addition, we would like to plot the pairs of experiments together in order to better compare the differences and similarities between experiments with glass and sand detachments.**

Page 14 line 34f: I suspect you will publish your data with GFZ Data Services in the framework of EPOS. I so, the sentence should be "Data. . .. Have been uploaded to the GFZ Repository published open access in McBeck et al. (2018)".

**See #20 above.**

References:

Adam, J., J.L. Urai, B. Wieneke, O. Oncken, K. Pfeiffer, N. Kukowski, J. Lohrmann, S. Hoth, W. van der Zee, J. Schmatz, (2005), Shear localisation and strain distribu- tion during tectonic faulting - New insights from granular-flow experiments and high- resolution optical image correlation techniques, Journal of Structural Geology, 27(2), 183-301,doi:10.1016/j.jsg.2004.08.008.

Ritter, M., K. Leever, M. Rosenau, and O. Oncken (2016): Scaling the Sand Box - Mechanical (Dis-) Similarities of Granular Materials and Brittle Rock, J. Geophys. Res - Solid Earth, doi.: 10.1002/2016JB012915.

Ritter, M.C., M. Rosenau, O. Oncken (2018a): Growing Faults in the Lab: Insights into the Scale Dependence of the Fault Zone Evolution Process, Tectonics, doi: 10.1002/2017TC004787

Ritter, M.C., T. Santimano, M. Rosenau, K. Leever, O. Oncken (2018b): Sand- box rheometry: Co-evolution of stress and strain in Riedel– and Critical Wedge– experiments, Tectonophysics, Volume 722, 2 January 2018, Pages 400-409, ISSN 0040-1951, doi: 10.1016/j.tecto.2017.11.018.

—————————end of review—————————

---

## Author Response (AR2)

*Response to Revisions*

**Dear Editor van Dinther,**

**We would be happy to add the acknowledgment of the helpful revisions of Dr. Rosenau and Dr. Bauville.**

**Sincerely,**
**Jessica McBeck**